# In Vitro and In Silico Analyses Explore the Role of Flavonoid Classes in the Antiviral Activity of Plant Extracts Against the Dengue Virus

**DOI:** 10.3390/molecules30234566

**Published:** 2025-11-27

**Authors:** Sindi A. Velandia, Elena E. Stashenko, Elizabeth Quintero-Rueda, Sergio Conde-Ocazionez, Lady J. Sierra, Raquel E. Ocazionez

**Affiliations:** 1Research Center for Chromatography and Mass Spectrometry (CROM-MASS), Universidad Industrial de Santander, Carrera 27, Calle 9, Bucaramanga 680002, Colombia; sindi.velandia@correo.uis.edu.co (S.A.V.); elena@tucan.uis.edu.co (E.E.S.); elizabeth.quintero@correo.uis.edu.co (E.Q.-R.); ladyjsierra@gmail.com (L.J.S.); 2Netherlands Institute for Neuroscience, Royal Netherlands Academy of Arts and Sciences, UMC Location University of Amsterdam, 1105 BA Amsterdam, The Netherlands; sacount2@gmail.com

**Keywords:** dengue virus, antiviral activity, in silico analysis, *Scutellaria coccinea*, *Lippia alba*, *Lippia origanoides*

## Abstract

This study examined the relationship between flavonoid content and the antiviral effects of plant extracts against the dengue virus (DENV). Fourteen extracts from medicinal plants cultivated in Colombia, which were prepared by ultrasonic-assisted solvent extraction (UAE) and supercritical fluid extraction (SFE) were included. UHPLC/ESI-Q-Orbitrap-MS analysis identified forty-six flavonoids. Antiviral effect on viral adsorption was evaluated using the DENV-CPE-based assay. UAE extracts of *Scutellaria coccinea*, *Scutellaria incarnata*, and *Lippia alba* contained higher amounts of flavonoid glycosides (from 97.0% to 87.9%) than aglycones, and showed antiviral effect (IC_50_: 3.0 to 65 µg/mL; SI: 0.4 to 71). In contrast, UAE and SFE extracts from *Lippia origanoides* had higher content of flavonoid aglycones (41.7% to 93.4%) than glycosides (0.0 to 58.3%) and showed no antiviral effect. Cluster and one-way ANOVA analyses revealed a positive correlation between increased levels of flavone glycosides in the extract and a strong antiviral effect. Docking analyses (AutoDock Vina) revealed that the flavonoid glycosides exhibited a higher binding affinity for the target proteins (E, Gas6-Axl, clathrin, and dynamin) than the aglycones did. This study establishes a scientific basis for using extracts rich in flavonoid glycosides, particularly flavones, as starting points for developing plant-based therapies to treat dengue.

## 1. Introduction

Dengue virus (DENV) is transmitted to humans by *Aedes* species mosquitoes, which are prevalent in tropical and subtropical regions worldwide. There are four serotypes of the virus: DENV-1, -2, -3 and -4 [1]. DENV can cause a range of symptoms, from an asymptomatic infection to dengue fever and severe dengue shock syndrome. Dengue is highly prevalent in South and Southeast Asia, where it accounts for nearly 70% of the world’s cases. These regions are considered epicenters of the disease [2]. The Caribbean, Central America, and South America are also heavily affected. The number of dengue cases reported to the World Health Organization has increased over eight-fold in the last two decades, from 2.4 million in 2010 and 4.2 million in 2019 [2]. The reported deaths between 2000 and 2015 increased from 960 to 4032. According to the Pan American Health Organization, dengue fever increased significantly in the Americas in 2024 rising by 232% compared to 2023 and 421% compared to the average of the previous five years [3]. Dengue has been identified as a significant and persistent public health concern in Colombia, which is among the most affected countries in the Americas. From 2012 to 2020, annual incidence rate ranged from 90.7 to 476.2 per 100,000 population, with the most recent outbreak reported in 2019 (465.9 per 100,000 population) [4].

There are currently no specific antiviral treatments for dengue, so therapy is primarily supportive [5]. Although two vaccines have been approved in some countries, their use is constrained due to safety concerns and their limited effectiveness against all DENV serotypes [6]. Dengue is a complex disease, making the discovery of effective therapies challenging. Increasing evidence suggests that herbal medicines have the potential as sources for clinical dengue treatments [7]. Administration of such medicines after virus exposure has shown to reduce the risk of severe dengue [7,8]. Thus, analyzing plant extracts has become an important strategy to discover and develop alternative therapies for dengue.

Plant-derived extracts prepared from a variety of species cultivated in different countries have demonstrated antiviral properties against human viruses, including DENV [7,8,9]. Their antiviral effect varies according to the plant species, chemotype, and phenological stages. The chemical composition of plant extracts depends on both the species and the extraction technique used which, in turn, affects their antiviral activity [10,11]. Diverse secondary metabolites, especially phenolic compounds such as flavonoids, have shown the potential to inhibit a variety of human pathogenic viruses, including influenza H1N1, HIV, and Zika [9,10,11]. Further research is necessary to fully elucidate chemical profiles that explain the anti-DENV activity of plant extracts. Studies demonstrate that plant-derived flavonoids have antiviral properties and can inhibit DENV by interfering with viral replication, entry, or specific viral proteins [12,13]. Several flavonoids derived from plants used in traditional medicine in Asia and Africa countries, such as quercetin, baicalin, baicalein, naringin, and catechin, have shown promising anti-DENV effects in vitro. Traditional Colombian herbal remedies are widely used to alleviate dengue symptoms, although few plants have been systematically studied for their therapeutic potential [14]. Research on the anti-DENV activity of plant extracts contributes to discovering alternative therapies to prevent severe dengue.

The genus *Scutellaria* (Lamiaceae) includes about 400 species widely distributed across America, Europe and Asia [15]. *S. baicalensis* is recognized as a promising natural source of compounds for treating viral diseases [16]. The genus *Lippia* (Verbenaceae) comprises 150 species, primarily found in the Americas, tropical Africa and India [17]. *L. alba* (Mill.) N.E. Brown was the first plant approved by the French Medicines Agency for inclusion in the French Pharmacopoeia [18]. *L. origanoides* Kunth was included in the *Formulário de Fitoterápicos da Farmacopéia Brasileira* due to its strong antibacterial properties [19]. *L. alba* and *L. origanoides* are traditionally used to treat viral infections such as influenza, measles and gastroenteritis [20,21]. Extracts prepared from *S. baicalensis* and *L. origanoides* have demonstrated inhibitory effect on the replication of DENV and other enveloped viruses [22,23]. We have investigated biological activities of both *Lippia* species and *Scutellaria* species cultivated in Colombia, and their extract chemical composition has been documented [23,24,25,26,27]. This study investigated the in vitro anti-DENV activity and the influence of flavonoid content in fourteen extracts prepared three species of *Scutellaria*, two chemotypes of *L. alba*, and three chemotypes of *L. origanoides* from Colombia. Furthermore, in silico analyses were performed to hypothesize how the flavonoids in the extracts might interfere with the entry of DENV into Vero cells.

## 2. Results

### 2.1. Extracts

Fourteen extracts from five plant species were selected for the study: *Scutellaria coccinea*, *Scutellaria incarnata*, *Scutellaria ventenatii* × *incarnata* (hybrid), *Lippia. alba*, and *Lippia origanoides*. Eleven extracts were prepared using ultrasound-assisted extraction (UAE) at different temperatures (50 °C and 47 °C) and times (60, 23, and 15 min). Three extracts from *L. origanoides* were prepared using supercritical fluid extraction (SFE). The characteristics of the extracts are summarized in Table 1 and are categorized by plant species, chemotype, and extraction technique. The codes for the extracts are assigned based on the genus (first letter), species (second letter), and chemotype (third letter) of the plant; the extraction technique (UA or SFE) used; and the sample number (1 or 2). Eight UAE extracts were prepared in this study: SiUA-2, SviUA, LacUA, LaiUA, LopUA-2, LocUA-1, LocUA-2, and LotUA. The remaining six extracts were prepared and chemically characterized in previous studies [23,24,25].

### 2.2. Cytotoxicity

Two different protocols of the crystal violet assay were used. One protocol determined the non-cytotoxic concentration of the extract to be used in the antiviral assay. This involved treating cells with extract for 1.5 h, which corresponded to the time allowed for viral replication. No cytotoxicity was observed for any of the extracts at concentrations ranging from 3.12 to 100 µg/mL (Figure 1A). The other protocol determined the 50% cytotoxicity concentration (CC_50_) of the extract by treating the cells for 72 h, in accordance with the standard test protocol for measuring the cytotoxicity of plant material (Figure 1B). Extracts from *Scutellaria* exhibited CC_50_ values ranging from 23 µg/mL to 213 µg/mL, for *S. ventenatti* + *incarnata* (SviUA) and *S. coccinea* (ScUA), respectively. Extracts from *L. alba*, LaiUA and LacUA, exhibited CC_50_ values of 65 µg/mL and 75 µg/mL, respectively. Six extracts from *L. origanoides* showed no cytotoxicity at the maximum concentration tested, suggesting CC_50_ values greater than 250 µg/mL; and the remaining two extracts from phellandrene (LopUA-2) and carvacrol (LocUA-2) chemotypes exhibited CC_50_ values of 117 µg/mL and 143 µg/mL, respectively.

### 2.3. Antiviral Effect

Virus-induced cytopathic effect (CPE) is a surrogate measure of virus replication in vitro; therefore, lower DENV-induced CPE indicates stronger antiviral activity [28]. The extracts were tested at six nontoxic concentrations to evaluate their effect on CPE caused by DENV-1 and DENV-2 after treatment during virus adsorption to Vero cells. The presence of DENV in untreated cells was confirmed by detection of non-structural protein 1 (NS1 = 75 ± 1.97 PanBio units) using an ELISA kit. Antiviral effects were classified as: (i) strong, greater than 50% reduction in CPE of both virus serotypes with IC_50_ values lower than 55 µg/mL; (ii) weak, IC_50_ ≤ 55 µg/mL for only one serotype or IC_50_ between 55 and 100 µg/mL for one or both serotypes; and (iii) inactive, no significant CPE reduction in either serotype. Selectivity index (SI) was included in the analysis. An extract with IC_50_ ≤ 55 µg/mL and SI ≥ 3.0 was considered a promising anti-DENV sample. The extracts exhibited variable anti-DENV effect and selectivity (Figure 2 and Table 2).

Extracts from *S. coccinea* (ScUA) and *S. incarnata* (SiUA-1) exhibited strong anti-DENV effect. ScUA showed IC_50_ values < 53 µg/mL and SI values > 4.0, indicating a promising anti-DENV sample. SiUA-1 showed IC_50_ values < 40 µg/mL, but SI values < 3.0, suggesting reduced antiviral promise. The other *S. incarnata* extract (SiUA-2) was inactive; significant reduction in the virus-induced CPE of both serotypes was not detected. The *S. ventenatti *+* incarnata* (SviUA) extract exhibited a weak antiviral effect, reduced DENV-2 at IC_50_ of 65 µg/mL (SI of 0.4), but not DENV-1. Extracts from *L. alba* exhibited weak antiviral effect. LaiUA (citral chemotype) reduced the replication of both virus serotypes, but at IC_50_ of 81 µg/mL against DENV-1. LacUA (carvone chemotype) showed antiviral promise against DENV-2 (IC_50_ of 19 µg/mL and SI of 3.9), but was inactive against DENV-1. All eight extracts from *L. origanoides* chemotypes were inactive against both virus serotypes.

### 2.4. UHPLC/ESI-Q-Orbitrap-MS Analysis of the Extracts Prepared in This Study

The analysis based on retention times (t_R_) and fragmentation patterns, identified only compounds belonging to the flavonoid class, including glycosides (n = 19), aglycones (n = 19) and methylated (n = 3) (Table 3 and Table 4).

Extracts from species of *Scutellaria* (UAE, 60 min at 50 °C) had higher concentrations of flavonoid glycosides than aglycones. Nine flavonoids were detected in *S. incarnata* (SiUA-2), with glycosides accounting for 64.7% (55–85 mg/g) of the total content. The predominant flavonoids were dihydrobaicalein-glucuronide, scutellarin, baicalin, baicalein, and wogonin. The flavonoid content of the *S. ventenatti* + *S. incarnata* (SviUA) extract was similar to that of SiUA-2, except for the absence of dihydrobaicalein-glucuronide and dihydrobaicalein. Flavonoid glycosides accounted for 55% (32/58 mg/g) of the SviUA content.

*L. alba* extracts (UAE, 60 min at 50 °C) had high flavonoid glycosides content, exceeding levels reported in extracts of the same species cultivated in Brazil [33], which may be attributed to variations in extraction methodology and cultivation practice of the plant. In the LaiUA (citral chemotype) extract were identified eight flavonoid glycosides accounting for 90.7% (16.6/18.3 mg/g) of total content. Tricin-7-diglucuronide, chrysoeriol-7-diglucuronide, chrysoeriol-7-glucuronide, tricin-glucuronide, and apigenin-7-glucuronide were predominant. Seven flavonoid aglycones were also identified, five of which were below detection limits. The LacUA (carvonal chemotype) extract exhibited a comparable profile, yet with higher amounts of flavonoid glycosides (97.1%, 20.7/21.3 mg/g).

*L. origanoides* extracts had higher flavonoid aglycone content than the other test extracts. Notable differences were observed in extracts from the carvacrol chemotype. LocUA-1 (UAE, 23 min at 47 °C) contained twenty-four flavonoids, 82.3% (11.2/13.6 mg/g) of which were aglycones. Eriodictyol, naringenin, luteolin, quercetin, and cirsimaritin were predominant. The LocUA-2 extract (UAE, 60 min at 50 °C) contained twelve flavonoids in higher amounts (259 mg/g vs. 13.6 mg/g), with aglycones accounting for 41.7%. The predominant flavonoids were eriodictyol, eriodictyol-7-glucoside, and luteolin-7-glucoside. The thymol chemotype LotUA extract (UAE, 23 min at 47 °C) contained twenty-three flavonoids. Aglycones accounted for 73.1% (42/57.4 mg/g), with eriodictyol, quercetin, and naringenin as the predominant. The phellandrene chemotype LopSE2 (UAE, 60 min at 50 °C) extract contained fourteen flavonoids, with aglycones accounting for 61.5% (228/371 mg/g) with eriodictyol, pinocembrin, and galangin as the predominant.

### 2.5. Relationship Between Anti-DENV Effect and the Flavonoid Content

Fourteen extracts listed in Table 1 were included in the analysis. The chemical compositions of six extracts that were analyzed in previous studies are shown in Appendix A. The relationship between antiviral effect and the flavonoid content was investigated to test the hypothesis that extracts exhibiting anti-DENV effect (strong or weak) possess distinct chemical profiles compared to inactive extracts. Table 5 compares flavonoid classes identified in the fourteen extracts.

Extracts with the strongest anti-DENV effects (ScUA and SiUA-1) contained high amounts of flavonoids (277.8 mg/g and 492.9 mg/g, respectively). Of these flavonoids, 83.4% and 86.1%, respectively, corresponded to flavone glycosides, with baicalin, dihydrobaicalein-glucuronide, and scutellarin being the predominant flavones. ScUA and SiUA-1 also had small amounts of flavone aglycones, 10.3% and 5%, respectively. Additionally, the two extracts contained non-flavonoid compounds (6.4% and 10.5%), including verbascoside and umbelliferone-hexoside-pentoside (Appendix A). In contrast, extracts with weak anti-DENV effects (SviUA, LacUA, and LaiUA) had significantly lower amounts of flavonoid (18.3 mg/g to 57.9 mg/g), though most of these were flavone glycosides (55.5% to 97% mg/g). Among the extracts with anti-DENV effect, SviUA contained a significantly higher concentration (44.5%) of flavonoid aglycones, all of them belonging to the flavone class. The *S. incarnata* SiUA-2 extract, lacking anti-DENV effect, had a chemical profile comparable to SviUA, except for the lack of dihydrobaicalein.

Extracts from *L. origanoides* lacking anti-DENV effect, except LopUA-1, had the lowest concentrations of flavone glycosides (0 to 24.7% vs. 64.8% to 97%) but the highest concentrations of aglycones (61.5% to 98.1% vs. 3.0% to 44.5%) compared to the other six extracts. The aglycones present were particularly flavanones and flavanols/flavanonols, which were absent in extracts that showed anti-DENV effects. Furthermore, UAE *L. origanoides* extracts exhibited higher levels of flavonoid glycosides non-flavones (14.7% to 52.7 vs. 0 to 4.6%), while SFE extracts lacked flavonoid glycosides and contained small amounts of methylated flavonoids. The LopUA-1 extract (phellandrene chemotype) had higher levels of flavonoid glycosides (77.6%) than aglycones (22%) compared to other *L. origanoides* extracts.

A Kohonen’s self-organized map algorithm, using percentages of flavonoid classes as input, grouped the fourteen extracts into two clusters (Figure 3A). Cluster 1 included five anti-DENV active extracts (*Scutellaria* and *L. alba*) along with the inactive *S. incarnata* extract (SiUA-2). These extracts shared the highest flavone glycosides content and lowest flavonoid aglycones. Cluster 2 grouped seven inactive *L. origanoides* extracts marked by elevated flavonoid aglycone levels. The LopUA-1 extract from *L. origanoides* was not included in cluster 2, as would have been expected, due to its distinct chemical profile. Figure 3B,C show that CPE values for DENV-1 and DENV-2 were significantly lower in cluster 1 compared to cluster 2 (Wilcoxon test, *p* < 0.01). This suggests a plausible relationship between the anti-DENV activity and a flavonoid profile characterized by a higher content of flavonoid glycosides, especially flavones, relative to flavonoid aglycones.

### 2.6. Molecular Interactions Between Flavonoids and DENV and Vero Cell Proteins

DENV infection of Vero cells is initiated by the interaction between its envelope (E) protein and specific cell surface receptors, including Axl receptor tyrosine kinase [34]. The virus enters cells via a non-classical endocytic pathway mediated by the clathrin and dynamin proteins [35,36,37]. To explore the role of flavonoids in the anti-DENV effect, an AutoDock Vina 1.5.6 software was used to dock forty-six flavonoids found in the fourteen extracts to the DENV-E, Gas6-Axl complex, clathrin, and dynamin. Binding energy values of ≤−7.55 kcal/mol indicate binding affinity between ligand and target. Figure 4 shows the relative concentrations of flavonoids in the extracts and their binding affinities to proteins. As expected, the analysis revealed varied binding affinities to targets. Generally, glycosides exhibited higher binding affinities (energy ranging from −10.22 to −8.01 kcal/mol) compared to aglycones (−8.23 to −7.90 kcal/mol), a difference that can be influenced by the presence of sugar. Most flavonoid glycosides exhibiting the lowest binding energies were the most abundant in extracts with anti-DENV effect.

The DENV-E protein structure consists of three domains [38]. DI and DII domains form the DI/DII hinge region containing a conserved fusion loop. Conformational changes in this region lead to the fusion of the viral and cell endosomal membranes [35]. Nineteen glycosides (−9.16 to −7.72 kcal/mol) and fourteen aglycones (−8.06 to −7.55 kcal/mol) were predicted to bind to E. As for flavonoids found in extracts with antiviral effect, all flavone glycosides exhibited affinity (−8.78 to −7.72 kcal/mol) for E, except schaftoside. The lowest binding energies were exhibited by chrysoeriol-7-diglucuronide, chrysoeriol-7-glucuronide, luteolin-7-glucuronide, apigenin-7-glucuronide, and tricin-7-diglucuronide, all of which are predominant in *L. alba* extracts. All flavonoid aglycones, except cirsimaritin, exhibited affinity for E; the predominant baicalein, dihydrobaicalein, and apigenin showed the lowest energies (−7.96 to −7.84 kcal/mol). As for flavonoids present in extracts lacking antiviral effect (*L. origanoides*), 50% glycosides non-flavones (−8.08 to −7.73 kcal/mol) and 50% aglycones (−8.05 to −7.73 kcal/mol) exhibited affinity for E. Eriodyctiol-7-glucoside, eriodyctiol, and pinocembrin had the lowest binding energies. Flavonoids formed hydrogen bonds with amino acid residues at three consensus binding sites (Figure 5). Most glycosides were accommodated within the DII domain closer to the dimer interface (A/B), with the remaining glycosides located within the interface DIII/DI domains closer to the fusion loop. Most aglycones were accommodated within the detergent β-octyl glucoside (βOG) site in the hinge region, with the remaining aglycones located within the DII (A/B) interface.

DENV can bind to the Axl receptor via the Gas6 protein, which facilitates virus entry by bridging the viral envelope’s phosphatidylserine to Axl [35,36]. Gas6 has two domains (Lg1 and Lg2) that interact with Axl domains (Ig1 and Ig2). Sixteen flavonoid glycosides were predicted to bind to Gas6 (−8.66 to −7.56 kcal/mol), whereas no aglycones (−7.52 to −6.83 kcal/mol) exhibited binding affinity. All flavone glycosides found in anti-DENV extracts exhibited affinity (−8.69 to −7.56 kcal/mol) for Gas6, except for schaftoside. The lowest binding energies were exhibited by chrysoeriol-7-diglucuronide, tricin-7-diglucuronide, luteolin-7-glucuronide, and luteolin-7-glucoside. In contrast, only two (eriodyctiol-7-glucoside and eriodyctiol-rhamnoside) from six glycoside non-flavones found in inactive extracts were predicted to bind to Gas6 (−7.77 kcal/mol and −7.76 kcal/mol, respectively). Flavonoid glycosides formed hydrogen bonds with amino acid residues at five consensus binding sites (Figure 6). Most flavonoids were located within the Gas6-Lg2/Lg1 interface of each monomer. The remaining flavonoids were accommodated within the Gas6-Lg1 domain, and the Gas6-Lg1/Axl interface of each monomer.

Dynamin possesses a GTPase domain that binds to and hydrolyzes guanosine-5′-triphosphate (GTP), causing a conformational change crucial for its function in DENV and cell membranes fusion [37]. Small molecules targeting GTPase can interfere with virus internalization [37,39]. All twenty-three flavonoid glycosides (−9.96 to −7.68 kcal/mol), except quercetin-3-glucoside, were predicted to bind to GTPase. The glycosides with the lowest binding energies were chrysoeriol-7-diglucuronide, baicalin, isoliquiritin, apigenin-7-glucoside, and chrysoeriol-7-glucuronide. These flavonoids are found in anti-DENV extracts but not in inactive *L. origanoides* extracts. Seventeen aglycones exhibited binding affinity for GTPase (−8.26 to −7.73 kcal/mol). Baicalein, dihydrobaicalein, luteolin, and scutellarein had the lowest binding energies. The flavonoids formed hydrogen bonds with amino acid residues of five consensus binding sites (Figure 7). Most glycosides were accommodated at the G4 loop extending to the switch1 loop (G4-switch1 loops), while others were accommodated within the P-loop and a few at the switch1 or switch2, closer to the P-loop (switch1-P-loops and switch2-P-loops). Most aglycones were accommodated at the switch2-P-loops and G4-switch1-loops.

Clathrin is a transport protein whose N-terminal domain (CTD) has various peptide-binding sites, including the PWDLW motif (W-box) [40]. CTD has been proposed as a target for antiviral agents that hinder viral entry into the cell [40]. All of the flavonoid glycosides were predicted (−10.01 to −8.08 kcal/mol) to bind to CTD, except for quercetin-3-glucoside. The lowest binding energies were observed for chrysoeriol-7-diglucuronide, triicin-7-diglucuronide, apigenin-7-glucoside, baicalin, and scutellarin. All flavonoid aglycones except wogonin exhibited binding affinity for CTD (−8.22 to −7.75 kcal/mol). Luteolin, eriodictyol, hesperetin, nepetin, and baicalein had the lowest binding energies. Both glycosides and aglycones formed hydrogen bonds with amino acid residues in the W-box motif (Figure 8).

Molecular docking results for the predominant flavonoids of extracts from *S. coccinea*, *S. incarnata*, *S. ventenatti* + *incarnata*, and *L. alba* are shown in Table 6. Results for the other flavonoids are shown in Appendix A.

## 3. Discussion

This study analyzed fourteen plant extracts from *Scutellaria* and *Lippia* species cultivated in Colombia that were prepared under varying experimental conditions. Analysis by UHPLC/ESI-Q-Orbitrap-MS revealed that flavonoids represented 95–100% of the content in all test extracts. UAE-extracts exhibited significant differences in their chemical profiles. Retention of flavonoid glycosides was greater with shorter extraction times (5 or 23 min) than with longer times (15 or 60 min). This trend was evident in comparisons between SiUA-1 vs. SiUA-2 as well as LopUA-1 vs. LopUA-2 extracts. The reduction in flavonoid glycosides by prolonged extraction is likely due to excessive sonication that facilitate thermal, oxidative, and mechanical degradation, which particularly affect thermolabile glycosides and phenolic compounds [41,42]. Time-dependent variation in the extraction process critically influences the chemical profiles, and ultimately, the abundance of bioactive compounds [41]. SFE is a highly efficient technique for extracting aglycone-rich fractions with limited recovery of water-soluble or glycosylated phytochemicals [43]. The SFE extracts of *L. origanoides* contained high levels of flavonoid aglycones and lacked glycosides, reflecting the principle that nonpolar CO_2_ extraction selectively enriches lipophilic compounds while excluding polar glycosides [44]. Chemical profiles of the extracts analyzed in this study differ from those reported for extracts of the same plant [21,29,30,31,32,33]. These variations can be explained by differences in growth stage and plant part, cultivation practice, and preparation technique.

The potential of crude plant extracts to inhibit DENV replication in vitro has been widely documented [7,8,9]. However, relationships between their chemical composition and anti-DENV effect remain to be elucidated. This study integrates UHPLC/ESI-Q-Orbitrap-MS data with CPE-DENV reduction data showing that higher contents of flavonoid glycosides, predominantly flavones, correlate with greater inhibitory effect on DENV replication in Vero cells. Moreover, increased aglycones are associated with lack of antiviral action, as evidenced by the extracts from *L. origanoides.* The distinction between *S. incarnata* extracts is notable: SiSE1 showed strong antiviral effect, whereas SiSE2 had none. SiUA-2 contained lower flavonoid glycoside content (64.8% vs. 88.6%) and sevenfold higher aglycones (35.2% vs. 5.0%). In a previous study [22], we compared the anti-DENV effects of LopUA-1 (77% glycosides) and LopSFE (no glycosides) extracts from *L. origanoides*. After treatment during DENV-1 adsorption on human hepatic cells, the level of viral NS1 protein was significantly reduced by LopSE1, but not by LopSFE. The flavonoid content is a determining factor in the antiviral action of plant extracts on viruses other than DENV [10,44]. Glycosylation may enhance flavonoid interactions with virus particles or cell surface receptors by increasing polarity or specific binding conformations relative to aglycones [44]. A study revealed that baicalin exhibited higher anti-DENV activity than baicalein in virus-infected cells [45]. Another study revealed that myricetin and its glycosides exhibited higher antiviral activity against HIV-1 virus compared to their aglycone counterparts [46].

While mechanisms underlying plant extract antiviral activity are yet to be fully elucidated, flavonoids are widely accepted as primary agents [12,13]. Certain flavonoids can interfere with viral adsorption to host cells through the attachment and subsequent inhibition of proteins and molecules involved in the virus endocytosis [46,47]. We evaluated the potential of the extracts to interfere with the adsorption of the virus to cells. The reduction in DENV-induced CPE suggests that flavonoids may block molecular components that promote virus-cell membrane interactions, a possibility supported by the in silico analysis results. We selected the E protein of DENV, which plays an important role in the process that allows the virus to enter cells. Flavonoid glycosides from extracts with an antiviral effect exhibited the strongest affinity for the amino acids of the E DII domain, which interacts with receptors to promote viral entry into cells. Additionally, flavonoid aglycones bind to the βOG binding site of E, which has been established as a target for developing dengue antivirals [48]. Many flavonoids have been identified as DENV-E protein ligands [13,49].

DENV uses a variety of cell receptors to gain entry into Vero cells [34,36]. The Gas6-Axl complex was selected for the in silico analysis. Flavonoid glycosides from anti-DENV active extracts showed good binding affinities to Gas6, suggesting potential biological interference with this receptor, whereas none of the flavonoid aglycones bound effectively. Flavonoid glycosides may have prevented DENV virions from adhering to Vero cells by blocking the PtdSer-Gas6 binding step. Disrupting the PtdSer-Gas6-Axl complex has been proposed as a potential therapeutic approach to inhibit DENV replication [35]. Flavonoids may also have entered the cells and affected the expression of Gas6 and Axl. Luteolin can downregulate the expression of these proteins in human cells [50].

Dynamin-dependent and clathrin-mediated endocytosis pathways are involved in DENV entry into Vero cells [34,36]. Targeting these pathways has been proposed as a strategy for developing antivirals [51,52]. The N-terminal domain (CTD) of clathrin and the GTPase domain of dynamin were selected for in silico analysis. Once again, flavonoid glycosides present in extracts with anti-DENV effect exhibited the best binding affinities with CTD and GTPase. Baicalein and scutellarin, which are abundant flavonoid aglycones in extracts from *Scutellaria* species, also showed good binding affinity to both proteins. The flavonoids may have destabilized the W-box domain of clathrin-CTD and reduced the capacity of dynamin to exchange GDP for GTP. The flavonoids may have affected the functionality of both proteins and consequently impacted the endocytosis process of DENV in Vero cells. Additionally, it is plausible that flavonoid treatment altered the fluidity of the viral envelope and cell membrane, affecting their fusion during endocytosis. Research has shown that flavonoids can increase the membrane fluidity by penetrating the hydrophobic bilayer core [53].

## 4. Materials and Methods

### 4.1. Reagents

HPLC-grade acetonitrile, HPLC-grade formic acid (FA), isopropanol (98%), ammonium formate (AF, ≥99%), LC/MS-grade methanol, and potassium persulfate (≥98%) were obtained from Merck (Darmstadt, Germany). Standard substances, including apigenin-7-glucoside, baicalin, galangin, quercetin-3-glucoside, taxifolin, acacetin, baicalein, luteolin, naringenin, nepetin, quercetin, sakuranetin, and salvigenin, were purchased from Sigma-Aldrich (St. Louis, MO, USA), ChemFaces (Wuhan, China), and Phytolab GmbH (Vestenbergsgreuth, Bavaria, Germany). Type I water was obtained from a Millipore Direct-QTM (Merck, Darmstadt, Germany) purification system. Eagle’s Minimum Essential Medium (MEM), fetal bovine serum (FBS), phosphate-buffered saline (PBS), and antibiotics were purchased from Gibco (Grand Island, NY, USA).

### 4.2. Viruses and Cell

Vero cells (ATCC^®^ CCL-81™), a monkey kidney epithelial cell line, were cultured in MEM supplemented with 100 units/mL penicillin-streptomycin mixture and 5% FBS, at 37 °C, in a 5% CO_2_ humidified incubator. BHK-21 cells (ATCC^®^ CCL-10™), a hamster fibroblast cell line, were cultured in MEM as aforementioned. DENV-1 (US/Hawaii/1944 strain) and DENV-2 (New Guinea C strain) viruses were propagated and titrated in serial 1-log dilutions to obtain a 50% tissue culture infectious dose (TCID50) in BHK-21 cells grown in 96-well plates.

### 4.3. Vegetal Material

*S. incarnata*, *S. ventenatti* × *incarnata*, *L. alba* (citral and carvone chemotypes), and *L. origanoides* (phellandrene, carvacrol, and thymol chemotypes) were cultivated in experimental plots at the Agroindustrial Pilot Complex of the National Center for the Agroindustrialization of Aromatic and Medicinal Tropical Plants (CENIVAM), Industrial University of Santander (UIS), Bucaramanga, Colombia. The Colombian Ministry of Environment and Sustainable Development authorized the use of plant material through Contract No. 270 for Access to Genetic Resources and Derived Products, signed with UIS. Voucher specimens were deposited in the UIS Herbarium. The aerial parts of the plants (leaves and stems) were harvested from February to November in both 2021 and 2023. They were dried in the dark at room temperature until a constant weight was achieved. Subsequently, the plant material was ground using a SM 100 RETSCH^®^ cutting mill (RETSCH^®^_,_ Haan, Germany) equipped with a 2.00 mm square-hole sieve. The powdered material was stored at room temperature (25 °C) in the absence of light. The mass lost during drying was determined gravimetrically.

### 4.4. Extraction Technique

Ultrasound-assisted extraction was used to prepare eight of the aforementioned extracts, as previously described [22,24]. Briefly, 100 g of plant material was mixed with 20 mL of 70% ethanol and water solution and then placed in an ultrasonic bath (Elma^®^ Elmasonic S15H, Singen, Germany). Extractions were conducted at 50 °C for 60 min, except for the *L. origanoides* thymol extract, which was conducted at 47 °C for 23 min (Table 1). The extracts were filtered using Whatman filter paper No. 1, after which the filtrate was extracted again with ethanol solution. The extracts were evaporated under vacuum using a Heidolph rotary evaporator, and then dried using a VirTis AdVantage Plus freeze dryer. Each extract (1 × 10^5^ µg/mL) was dissolved in 1% DMSO and stored at −20 °C.

### 4.5. UHPLC/ESI-Q-Orbitrap-MS Analysis

Analyses were performed in accordance with the protocol used in a previous study [24]. Eight UAE extracts were analyzed using an Ultimate Dionex^TM^ 3000 UHPLC system (Thermo Fisher Scientific, Bremen, Germany) coupled to a Q-Exactive Plus Orbitrap™ mass spectrometer (Thermo Fisher Scientific, Bremen, Germany), equipped with a heated electrospray ionization source (HESI-II) operated in positive-ion acquisition mode. Phenolic compounds were separated using a ZORBAX Eclipse XDB-C18 column (Agilent Technologies, Palo Alto, CA, USA), 50 mm, L × 2.1 mm, I.D. × 1.8 μm particle size, at 40 °C. The mobile phase consisted of 0.2% formic acid in water (A) and 0.2% formic acid in acetonitrile (B), delivered at 0.3 mL/min. The gradient started at 100% A, shifted to 100% B over 8 min, held for 4 min, returned to 100% A in 1 min, and was equilibrated for 3 min. The injection volume was 2 µL. Extracts were analyzed in positive-ion mode under the following parameters: electrospray ionization temperature, 350 °C; capillary temperature, 320 °C; capillary voltage, 3.5 kV; mass range *m*/*z* 80–1000; and higher-energy collisional dissociation (HCD) cell, 10–40 eV range. Data were processed using Thermo Xcalibur™ Roadmap (v3.1.66.10). Metabolites were identified by comparing their retention times (t_R_), accurate masses, isotopic ratios, and fragmentation patterns with those of reference standards and database entries [33,54]. Quantitative analysis of flavonoids was performed using calibration curves prepared with analytical standards of apigenin-7-glucoside, baicalin, galangin, quercetin-3-glucoside, taxifolin, acacetin, baicalein, luteolin, naringenin, nepetin, quercetin, sakuranetin, and salvigenin. Quantification was based on the peak areas obtained from extracted ion chromatograms, and results were expressed as milligrams of compound per gram of freeze-dried extract (mg/g) or as equivalents to baicalin, galangin, taxifolin, or baicalein. The limits of detection (LOD) ranged from 0.05 to 1.10 mg/L, depending on the compound. All analyses were performed in triplicate, and results are presented as mean ± standard deviation (SD).

### 4.6. Cytotoxicity Test

The crystal violet assay was used to evaluate the effect of the extracts on the viability of non-DENV-infected Vero cells [54]. Each extract was examined at seven concentrations (3.12 to 250 µg/mL) and with different treatment times. (A) Cells grown in 96-well plates were incubated with extract for 1.5 h, washed with PBS, and fresh culture medium (MEM; 2% FBS) was added. The plates were then incubated at 37 °C with 5% CO_2_, and cell viability was measured five days later. (B) Cells grown in 96-well plates were incubated in culture medium containing extract at 37 °C with 5% CO_2_, and cell viability was measured 72 h later. In both assays, untreated cells and cells treated with DMSO were run in parallel as negative and positive controls, respectively. Cell viability was measured by discarding the culture supernatant and adding a crystal violet solution (0.05%), followed by methanol. The plates were analyzed using an ELISA plate reader at a wavelength of 570 nm. Non-cytotoxic concentrations for the cells and CC_50_ values were calculated using nonlinear regression analysis with the was performed using GraphPad Prism version 10.0 for Windows, GraphPad Software, Boston, MA USA, www.graphpad.com.

### 4.7. Cytopathic Effect (CPE)-Based Antiviral Assay

The CPE-based assay was used to evaluate the antiviral effect of the test extracts, as previously described [54]. Confluent Vero cells monolayers were infected by adding DENV-1 or DENV-2 at a multiplicity of infection of 1 p.f.u./cell for 1.5 h, and in the presence of non-cytotoxic concentrations (3.12 to 100 µg/mL) of extract. After washing with PBS, the cells were incubated in fresh medium (MEM; 2% SBF) at 37 °C and 5% CO_2_ for five days to allow for virus replication. Virus-infected cells in the absence of extract (negative control) and virus-infected cells in the presence of SDS (positive control) were included. Viral CPE reduction was determined by measuring cell viability using the aforementioned crystal violet assay. Results of the screening were expressed as a percentage of the reduction in DENV CPE [(OD_570_ of infected treated cells − OD_570_ of negative control)/(OD_570_ of positive control − OD_570_ of negative control) × 100]. The concentration of extract that inhibited DENV-CPE by 50% (IC_50_) was calculated using nonlinear regression analysis (GraphPad Prism version 10.0 for Windows). Selectivity index (SI = CC_50_/IC_50_) values were calculated. Each extract was analyzed in triplicate in three independent assays.

### 4.8. Molecular Docking Analysis

Docking analyses were carried out using AutoDock Vina (Version 1.5.6, La Jolla, CA, USA), as previously described [54,55]. The 3D crystal structures of target proteins were retrieved from Research Collaboratory for Structural Bioinformatics protein data bank RCSB PDB (rcsb.org accessed on September 2024): DENV-2 E (ID: 10AN), Gas6-Axl receptor (ID: 2C5D), clathrin CDT domain (ID: 2XZG), and dynamin GTPase domain (PDB ID: 2X2E). The protein structures were cleaned, water molecules removed, Gasteiger charges computed, polar hydrogens added, and non-polar hydrogens merged. The structures of 46 flavonoids were retrieved from the PubChem database (https://pubchem.ncbi.nlm.nih.gov/, accessed on 15 September 2024), Table 6 and Appendix A show structures and their CIDs. The preparation of ligands after the energy minimization step started with detecting and choosing the root atom. Optimized protein and ligand structures were saved in the PDBQT format. Default parameters were used, and the search exhaustiveness parameter was set to 100. For each ligand, 27 docked conformations were generated using global docking simulations. Three simulations were performed for each ligand-protein pair using seeds 6, 12, and 18. The Vina output including the docking scores (Kcal/mol), upper and lower bound root mean square deviation (RMSD), and the number of hydrogen bonds and interacting residues were used to examine the best-docked conformations in the individual run. Discovery Studio Visualizer v21.1.0.20298 was used to visualize interacting residues between ligand and protein for structural analysis.

### 4.9. Statistical Analysis

The relationship between antiviral activity and the flavonoid content was analyzed using two independent parameters: the chemical composition, represented as the absolute percentage (%) of different flavonoid classes (Table 5), and the antiviral effect, DENV-CPE (%) as a unidimensional variable ranging from 0 to 100. The extracts were clustered according to their flavonoid content using an unsupervised Kohonen self-organizing map (SOM) [56], a type of unsupervised artificial neural network that groups samples by reducing the number of variables in the analysis, as previously described. Briefly, each extract was represented by a vector of chemical features and mapped onto a two-dimensional grid of ‘neurons’, where each neuron corresponds to a random vector of the same dimension as the input. During training, the algorithm identifies the neuron that most closely matches each extract vector based on distance metrics, then adjusts the weights of this and its neighboring neurons to further resemble the input. This process was repeated for 500 iterations, causing similar extracts to cluster near each other on the map, while more distinct samples separated into different regions of the grid. Consequently, extracts with similar chemical profiles cluster around the same neuron, whereas samples with dissimilar characteristics form distinct clusters. One-way ANOVA and Tukey–Kramer post hoc tests were used to compare DENV-CPE (%) values between clusters, adopting a significance level of 0.05. All analyses were performed using MATLAB^®^ R2021b (The MathWorks, Inc. (Torrance, CA, USA)).

## 5. Conclusions

This study reports for the first time the correlation between the antiviral efficacy of extracts from medicinal plants cultivated in Colombia and their flavonoid content. Integration of antiviral assays, UHPLC analysis, and molecular docking analysis revealed that the content in flavonoid glycosides is a key determinant of the extracts’ anti-DENV effect. Extracts rich in flavonoid glycosides, particularly flavones, demonstrated the greatest antiviral efficacy. In contrast, extracts with lower levels of glycosides or lacking them did not show antiviral activity. Docking analysis suggests that the flavonoids present in the extracts could block the interaction between DENV particles and the cell membrane via different mechanisms. Based on IC_50_ and SI values from the CPE-based antiviral assay, the UAE extract from *S. coccinea* can be classified as a prospective anti-DENV sample. This extract contains high concentrations of flavonoid glycosides with potential to inhibit DENV, and therefore could serve as a starting point for research on herbal medications for dengue treatment. There is considerable evidence demonstrating the potential of plants from the *Scutellaria* genus as a source for developing an antiviral agent against various pathogenic viruses, including dengue virus (DENV).

## Figures and Tables

**Figure 1 molecules-30-04566-f001:**
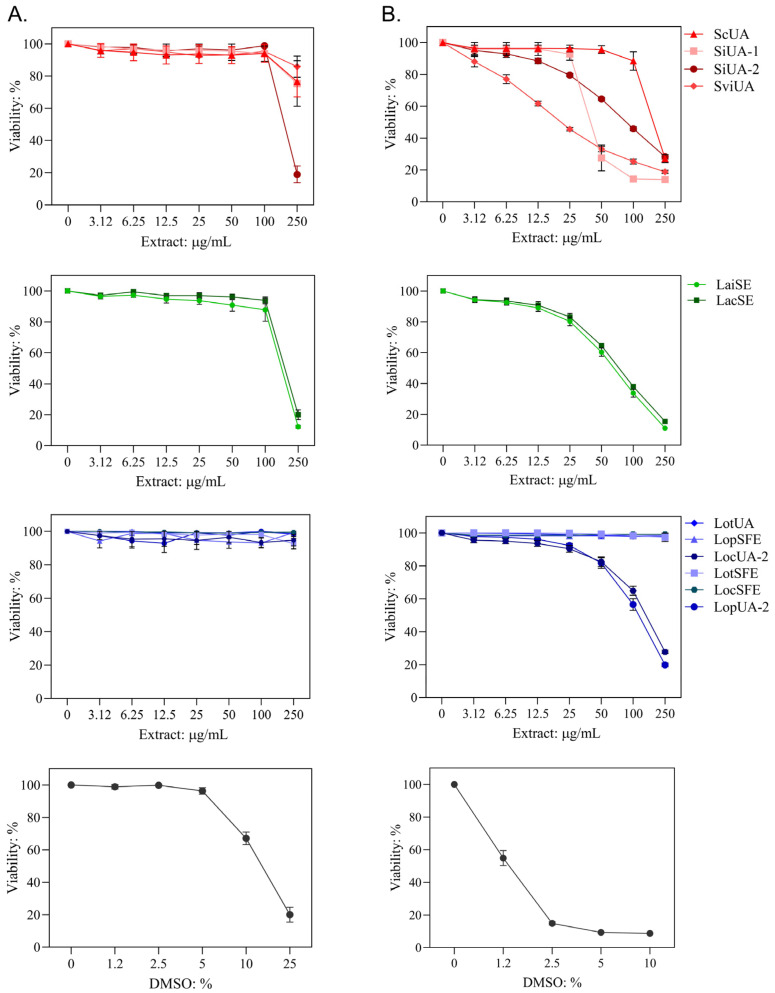
Cytotoxicity of the extracts in the crystal violet assay. (**A**) Vero cells were treated with extract for 1.5 h and after washing the cell viability was measured five days later. (**B**) Cells were cultured in medium containing extract and the cell viability was measured 72 h later. Extracts from *Scutellaria* in red: *S. coccinea* (ScUA), *S. incarnata* (SiUA-1 and SiUA-2) and *S. ventenatti* + *incarnata* (SviUA). Extracts from *L. alba* chemotypes in green: carvone (LacUA) and citral (LaiUA) chemotypes. Extracts from *L. origanoides* chemotypes in blue: phellandrene (LopUA-2, LopSFE), carvacrol (LocUA-1, LocUA-2, LocSFE) and thymol (LotUA, LotSFE) chemotypes. LopUA-1 and LocUA-1 extracts were not included and their dose responses were comparable to those of LotSFE. Dimethyl sulfoxide (DMSO) was used as a cytotoxic agent. Data are presented as mean ± SD of six measurements from three independent analyses.

**Figure 2 molecules-30-04566-f002:**
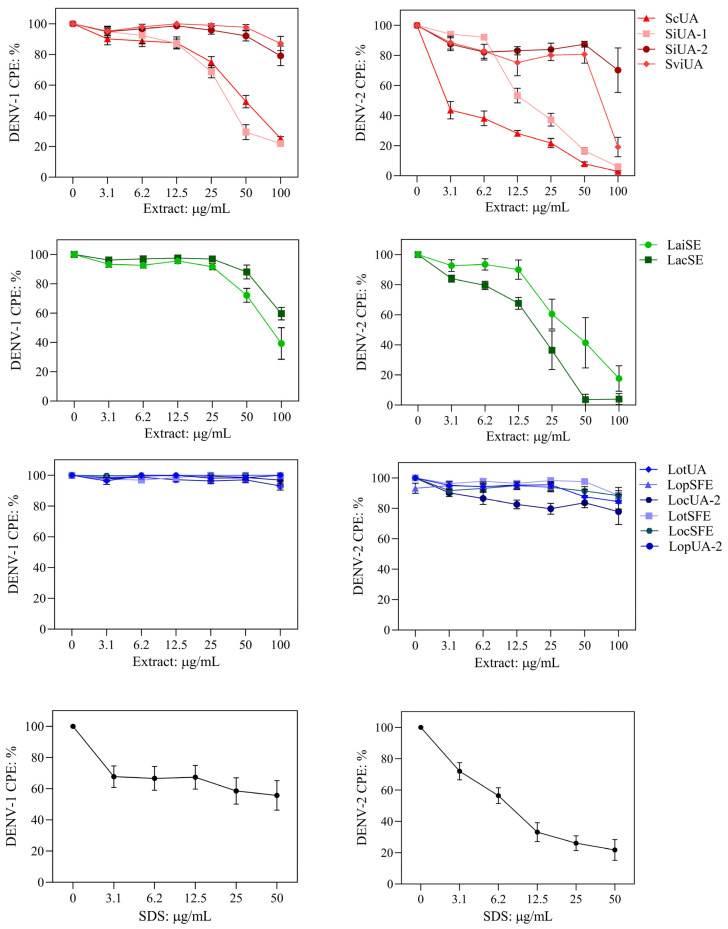
Antiviral effect of the extracts against dengue virus (DENV-1 and DENV-2) in the cytopathic effect (CPE)-based assay. Extracts from species of *Scutellaria* (red), *L. alba* chemotypes (green) and *L. origanoides* chemotypes (blue) as described in Figure 1. Extracts were added during virus adsorption to Vero cells. Quantification of crystal violet staining was performed to indirectly measure virus-induced CPE. DENV-CPE %: [(OD_570_ of virus-infected and extract-treated cells/OD_570_ of non-infected non-treated cells].: Sodium dodecyl sulfate (SDS) is an antiviral agent. Data are presented as mean ± SD of six measurements from three independent analyses.

**Figure 3 molecules-30-04566-f003:**
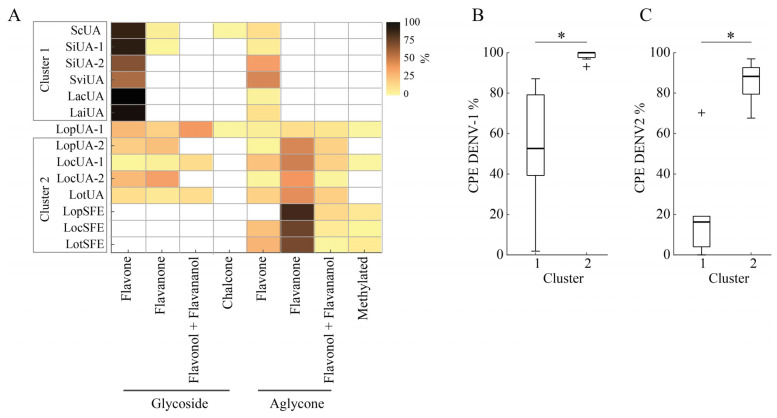
Relationship between flavonoid profiles and anti-DENV effects of the extracts. (**A**) Heatmap showing the content of flavonoid classes and revealing two clusters obtained by an unsupervised self-organizing map. (**B**,**C**) Comparison of antiviral activity (CPE, cytopathic effect percentage) of clusters against DENV-1 (* Wilcoxon test Z = −2.969; *p* = 0.003) and DENV-2 (* Wilcoxon test Z = −2.785; *p* = 0.0053). “+” symbols denote individual mild outliers falling outside the whiskers.

**Figure 4 molecules-30-04566-f004:**
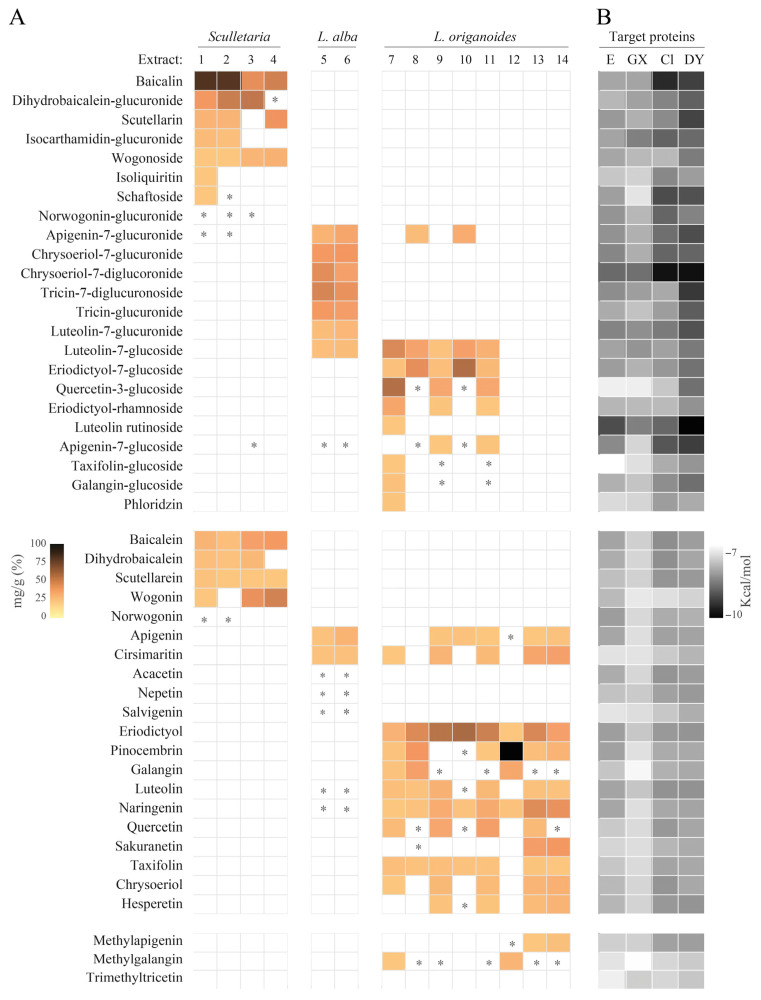
Heatmaps showing the abundance of each flavonoid and its binding affinity for target proteins. (**A**) Relative concentration (mg/g) of flavonoid in the extract (Table 4): 1, ScUA; 2, SiUA-1; 3, SiUA-2; 4, SviUA; 5, LacUA; 6, LaiUA; 7, LopUA-1; 8, LopUA-2; 9, LocUA-1; 10, LocUA-2; 11, LotUA; 11, LopSFE; 12, LocSFE; and 13, LotSFE. *Scutellaria* (1,2 and 4) and *L. alba* (6 and 7) exhibited anti-DENV effect, while the other extracts were inactive. * Low detection limit. (**B**) Binding affinities (kcal/mol) of flavonoids for target proteins: E, DENV-2 E envelope; GX, Gas6-Axl receptor; Cl clathrin N-terminal domain (2XZG); DY, dynamin GTPase domain.

**Figure 5 molecules-30-04566-f005:**
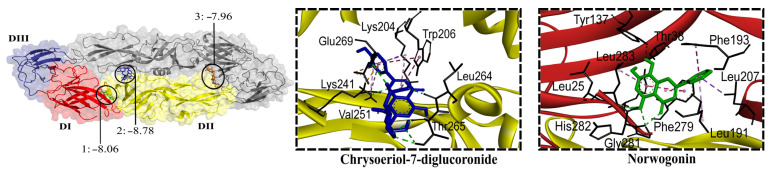
Contact preferences at the dimeric DENV-E protein (PDB ID: 1OAN) of representative flavonoids. The domains I, II and III of one monomer are colored red, yellow, and blue, respectively. Norwogonin (1) within the DI/DII hinge region around the βOG pocket; chrysoeriol-7-diglucoronide (2) within the interface of DII domains; and baicalin (3) within the DIII/DI interface. Binding free energies (kcal/mol) are shown.

**Figure 6 molecules-30-04566-f006:**
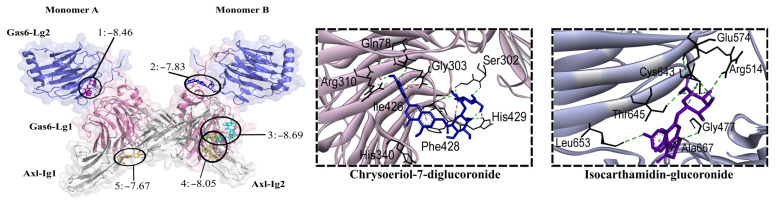
Contact preferences at the Gas6-Axl complex (PDB ID: 2C5D) of representative flavonoids. Gas6, Lg2 and Lg1 domains in blue and purple, respectively. Axl, Ig1 and Ig2 domains in gray. Isocarthamidin-glucuronide (1) and chrysoeriol-7-glucuronide (2) within the Gas6-Lg2/Lg1 interface; chrysoeriol-7-diglucuronide (3) within Gas6-Lg1 domain; tricin-7-diglucuronoside (4) within the Gas6-Lg1/Axl-Ig2 interface; and eriodyctiol-rhamnoside (5) within the Gas6-Lg1/Axl-Ig1 interface. Binding free energies (kcal/mol) are shown.

**Figure 7 molecules-30-04566-f007:**
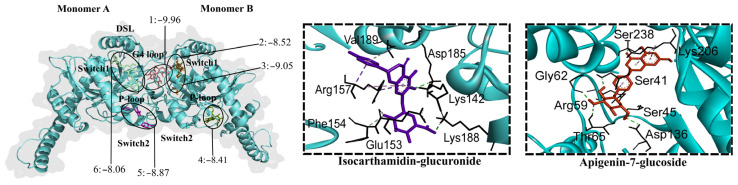
Contact preferences at the GTPase domain (PDB ID: 2X2E) of representative flavonoids. Chrysoeriol-7-diglucuronide (1) within the G4-switch1-loops; apigenin-7-glucoside (2) closer to the switch1-loop; luteolin-7-glucuronide (3) within switch1-P-loops; dihydrobaicalein-glucuronide (4) closer to the P-loop; isocarthamidin-glucuronide (5) within the switch2-P-loops; and tricin-glucuronide (6) within the switch1-loop. Binding free energies (kcal/mol) are shown.

**Figure 8 molecules-30-04566-f008:**
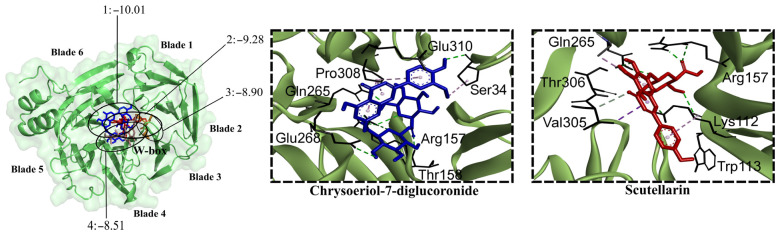
Contact preferences at the CTD-clathrin domain (PDB ID: 2XZG) of representative flavonoids. Chrysoeriol-7-diglucuronide (1), scutellarin (2), dihydrobaicalein-glucuronide (3), and wogonoside (4) within the W-box motif. Binding free energies (kcal/mol) are shown.

**Table 1 molecules-30-04566-t001:** Extracts included in the study.

Plant	Voucher	Extract Code	Extraction	
			Technique	Parameters
*Scutellaria coccinea* Kunth	UIS219784	ScUA	UAE	15 min; 50 °C
*Scutellaria incarnata* Vent	UIS219783	SiUA-1	UAE	5 min; 50 °C
		SiUA-2	UAE	60 min; 50 °C
*Scutellaria ventenatii* + *incarnata* (hybrid)	UIS219785	SviUA	UAE	60 min; 50 °C
*Lippia alba* (Mill) N.E. Brown	UIS22031 ^a^	LacUA	UAE	60 min; 50 °C
	UIS22002 ^b^	LaiUA	UAE	60 min; 50 °C
*Lippia origanoides* Kunth	UIS22035 ^c^	LopUA-1	UAE	23 min; 47 °C
		LopUA-2	UAE	60 min; 50 °C
		LopSFE	SFE	96 min; 307 bar
	UIS22034 ^d^	LocUA-1	UAE	23 min; 47 °C
		LocUA-2	UAE	60 min; 50 °C
		LocSFE	SFE	96 min; 307 bar
	UIS19799 ^e^	LotUA	UAE	23 min; 47 °C
		LotSFE	SFE	96 min; 307 bar

*Lippia* chemotypes are based on the major components of their essential oils. ^a^ Carvone, ^b^ citral, ^c^ phellandrene, ^d^ carvacrol, and ^e^ thymol. UAE, ultrasound-assisted solvent extraction. SFE, supercritical fluid extraction. The following extracts were prepared in previous studies: ScUA and SiUA-1 [24]; LopUA-1 [22]; and LopSFE, LocSFE and LotSFE [23].

**Table 2 molecules-30-04566-t002:** Antiviral effect of the extracts on DENV-infected Vero cells.

Extract Code	CC_50_: µg/mL	DENV-1			DENV-2			Effect
		CPE %	IC_50_: µg/mL	SI	CPE: %	IC_50_: µg/mL	SI	
ScUA ^a^	213 ± 1.1	25.3 ± 1.2	52 ± 1.3	4.1	2.9 ± 0.45	2.9 ± 1.3	71	Strong
SiUA-1 ^a^	50 ± 1.2	21.9 ± 1.4	39 ± 1.2	1.2	6.1 ± 0.97	17.9 ± 1.2	2.9	Strong
SiUA-2 ^a^	91 ± 1.05	79.1 ± 6.5	-	-	70.2 ± 14.7	-	-	Inactive
SviUA ^a^	23 ± 2.0	87.1 ± 4.7	-	-	19.1 ± 6.5	65.4 ± 1.2	0.4	Weak
LaiUA ^b^	65 ± 1.1	39.3 ± 10.8	81.8 ± 1.1	0.8	17.6 ± 8.4	37.5 ± 1.2	1.7	Weak
LacUA ^b^	75 ± 1.0	59.6 ± 4.3	-	-	3.9 ± 3.7	19.4 ± 1.1	3.9	Weak
LopUA-1 ^c^	>250.0	100	-	-	86.2 ± 3.7	-	-	Inactive
LopUA-2 ^c^	117 ± 1.0	93.0 ± 2.7	-	-	77.8 ± 8.4	-	-	Inactive
LopSFE ^c^	>250.0	100	-	-	93.9 ± 3.2	-	-	Inactive
LocUA-1 ^c^	>250.0	100	-	-	67.6 ± 11.2	-	-	Inactive
LocUA-2 ^c^	143 ± 1.1	100	-	-	88.2 ± 3.2	-	-	Inactive
LocSFE ^c^	>250.0	96.8 ± 2.5	-	-	96.2 ± 2.0	-	-	Inactive
LotUA ^c^	>250.0	100	-	-	84.3 ± 4.9	-	-	Inactive
LotSFE ^c^	>250.0	99.7 ± 0.2	-	-	88.5 ± 5.2	-	-	Inactive

^a^ *Scutellaria*, ^b^ *Lippia alba*, and ^c^ *Lippia origanoides* extracts. CC_50_ > 250 µg/mL was adopted for extracts that did not exhibit a 50% reduction in cell viability (Figure 1B). CPE %: percentage of virus-induced cytopathic effect after treatment with 100 µg/mL relative to the untreated control (100% CPE). SI, selectivity index (CC_50_/IC_50_).

**Table 3 molecules-30-04566-t003:** Flavonoids in the extracts prepared in the study identified by UHPLC/ESI-Q-Orbitrap-MS.

Name	Formula	Exp. Masses [M+H]^+^	Δ ppm	HCD, eV	Productions			References
					Fragment Type	Formula	(*m*/*z*)	
Apigenin-7-glucoside	C_21_H_20_O_10_	433.11287	0.11	30	[(M+H)−C_6_H_10_O_5_]^+^ [(M+H)−C_6_H_10_O_5_−H_2_O]^+^ [(M+H)−C_6_H_10_O_5_−CO]^+^ [(M+H)−C_6_H_10_O_5_−C_8_H_6_O]^+^ [(M+H)−C_6_H_10_O_5_−C_8_H_6_O_3_]^+^	C_15_H_11_O_5_ C_15_H_9_O_4_ C_14_H_11_O_4_ C_7_H_5_O_4_ C_7_H_5_O_2_	271.06012 253.04997 243.06558 153.01848 121.02852	[21,25]
Apigenin-7-glucuronide	C_21_H_18_O_11_	447.09056	0.54	20	[(M+H)−C_6_H_8_O_6_]^+^ [(M+H)−C_6_H_8_O_6_−C_8_H_6_]^+^	C_15_H_11_O_5_ C_7_H_5_O_5_	271.06033 169.01256	[21,25]
Baicalin	C_21_H_18_O_11_	447.09056	0.33	20	[(M+H)−C_6_H_8_O_6_]^+^ [(M+H)−C_6_H_8_O_6_−C_8_H_6_]^+^	C_15_H_11_O_5_ C_7_H_5_O_5_	271.06033 169.01256	[21,25]
Chrysoeriol-7-diglucuronide	C_28_H_28_O_18_	653.13484	0.50	20	[(M+H)−C_6_H_8_O_6_]^+ ^ [(M+H)−2C_6_H_8_O_6_]^+^	C_22_H_21_O_12 _ C_16_H_13_O_6_	477.10275 301.07066	[29,30]
Chrysoeriol-7-glucuronide	C_22_H_20_O_12_	477.10275	0.20	20	[(M+H)−C_6_H_8_O_6_]^+^	C_16_H_13_O_6_	301.07066	[29,30]
Dihydrobaicalein-glucuronide	C_21_H_20_O_11_	449.10602	0.44	10	[(M+H)−C_6_H_8_O_6_]^+^ [(M+H)−C_6_H_8_O_6_−C_8_H_8_]^+^ [(M+H)−C_6_H_8_O_6_−C_6_H_6_O_4_]^+^	C_15_H_13_O_5_ C_7_H_5_O_5_ C_9_H_7_O	273.07538 169.01341 131.04932	[23]
Eriodictyol-7-glucoside	C_21_H_22_O_11_	451.11327	0.48	10	[(M+H)−C_6_H_10_O_5_]^+^ [(M+H)−C_6_H_10_O_5_−H_2_O]^+^ [(M+H)−C_6_H_10_O_5_−C_8_H_8_O_2_]^+^	C_15_H_13_O_6_ C_15_H_11_O_5_ C_7_H_5_O_4_	289.07062 271.05969 153.01810	[29,31]
Luteolin-7-glucuronide	C_21_H_18_O_12_	463.08710	1.57	30	[(M+H)−C_6_H_8_O_6_]^+^ [(M+H)−C_6_H_8_O_6_−H_2_O]^+^	C_15_H_11_O_6_ C_15_H_9_O_5_	287.05501 269.04444	[29,31]
Eriodictyol-rhamnoside	C_21_H_22_O_10_	435.12845	0.28	30	[(M+H)−C_6_H_10_O_4_]^+^ [(M+H)−C_6_H_10_O_4_−H_2_O]^+^ [(M+H)−C_6_H_10_O_4_−H_2_O−C_6_H_4_O_2_]^+^ [(M+H)−C_6_H_10_O_4_−C_8_H_8_O_2_]^+^	C_15_H_13_O_6_ C_15_H_11_O_5_ C_9_H_7_O_3_ C_7_H_5_O_4_	289.07086 271.05937 163.03883 153.01846	[29]
Galangin-glucoside	C_21_H_20_O_10_	433.11287	0.11	20	[(M+H)−C_6_H_10_O_5_]^+^ [(M+H)−C_6_H_10_O_5_−C_2_H_2_O]^+^ [(M+H)−C_6_H_10_O_5_−C_6_H_4_O_2_]^+^ [(M+H)−C_6_H_10_O_5_−C_8_H_6_O]^+^	C_15_H_11_O_5_ C_13_H_9_O_4_ C_9_H_7_O_3_ C_7_H_5_O_4_	271.06012 229.04935 163.03894 153.01825	[20,29]
Luteolin-7-glucoside	C_21_H_20_O_11_	449.10791	1.07	20	[(M+H)−C_6_H_10_O_5_]^+^ [(M+H)−C_6_H_10_O_5_−C_2_H_2_O]^+^ [(M+H)−C_6_H_10_O_5_−C_6_H_4_O_2_]^+^ [(M+H)−C_6_H_10_O_5_−C_8_H_6_O_2_]^+^	C_15_H_11_O_6_ C_13_H_9_O_5_ C_9_H_7_O_4_ C_7_H_5_O_4_	287.05515 245.04386 179.03411 153.01846	[29,31]
Luteolin-rutinoside	C_27_H_30_O_15_	595.16524	0.86	10	[(M+H)−C_6_H_10_O_4_]^+^ [(M+H)−C_6_H_10_O_4_−C_6_H_10_O_5_]^+^ [(M+H)−C_6_H_10_O_4_−C_6_H_10_O_5_−C_8_H_6_O_2_]^+^ [(M+H)−C_6_H_10_O_4_−C_6_H_10_O_5_−C_8_H_6_O_3_]^+^	C_21_H_21_O_11_ C_15_H_11_O_6_ C_7_H_5_O_4_ C_7_H_5_O_3_	449.10864 287.05530 153.01832 137.02325	[29,31]
Phloridzin	C_21_H_24_O_10_	437.14386	0.82	10	[(M+H)−C_6_H_10_O_5_]^+^ [(M+H)−C_6_H_10_O_5_−H_2_O]^+^ [(M+H)−C_6_H_10_O_5_−C_7_H_6_O]^+^	C_15_H_15_O_5_ C_15_H_13_O_4_ C_8_H_9_O_4_	275.09131 257.08081 169.04961	[23,30]
Quercetin-3-glucoside	C_21_H_20_O_12_	465.10287	0.35	10	[(M+H)−C_6_H_10_O_5_]^+^ [(M+H)−C_6_H_10_O_5_−H_2_O]^+^ [(M+H)−C_6_H_10_O_5_−C_8_H_6_O_3_]^+^	C_15_H_11_O_7_ C_15_H_9_O_6_ C_7_H_5_O_4_	303.04971 285.03970 153.01811	[23]
Scutellarin	C_21_H_18_O_12_	463.08762	0.42	10	[(M+H)−C_6_H_8_O_6_]^+^ [(M+H)−C_6_H_8_O_6_−C_6_H_4_O_3_]^+^	C_15_H_11_O_6_ C_9_H_7_O_3_	287.05537 163.03885	[25]
Taxifolin-glucoside	C_21_H_22_O_12_	467.11829	0.24	20	[(M+H)−C_6_H_10_O_5_]^+^ [(M+H)−C_6_H_10_O_5_−H_2_O]^+^ [(M+H)−C_6_H_10_O_5_−C_8_H_8_O_3_]^+^	C_15_H_13_O_7_ C_15_H_11_O_6_ C_7_H_5_O_4_	305.06561 287.05496 153.01832	[23]
Tricin-7-diglucuronoside	C_29_H_30_O_19_	683.14540	0.28	30	[(M+H)−C_6_H_8_O_6_]^+^ [(M+H)−2C_6_H_8_O_6_]^+^	C_23_H_23_O_13 _ C_17_H_15_O_7_	507.11331 331.08122	[30,32]
Tricin-glucuronide	C_23_H_22_O_13_	507.11331	1.03	20	[(M+H)−C_6_H_8_O_6_]^+^	C_17_H_15_O_7_	331.08063	[30,32]
Wogonoside	C_22_H_20_O_11_	461.17773	0.47	20	[(M+H)−C_6_H_8_O_6_]^+^ [(M+H)−C_6_H_8_O_6_−CH_3_]^+•^	C_16_H_13_O_5_ C_15_H_10_O_5_	285.07535 270.05164	[25]
Acacetin	C_16_H_12_O_5_	285.07541	1.19	50	[(M+H)−CH_3_]^+•^ [(M+H)−CH_3_−CO]^+•^ [(M+H)−C_9_H_8_O]^+^ [(M+H)−C_8_H_6_O_4_]^+^	C_15_H_10_O_5_ C_14_H_10_O_4_ C_7_H_5_O_4_ C_8_H_7_O	270.05273 242.05754 153.01842 119.049228	[30,33]
Apigenin	C_15_H_10_O_5_	271.06042	0.82	40	[(M+H)−H_2_O]^+^ [(M+H)−CO]^+^ [(M+H)−C_8_H_6_O]^+^	C_15_H_9_O_4_ C_14_H_11_O_4_ C_7_H_5_O_4_	253.04997 243.06558 153.01848	[21,22]
Baicalein	C_15_H_10_O_5_	271.05969	1.50	60	[(M+H)−H_2_O]^+^ [(M+H)−H_2_O−CO]^+^ [(M+H)−C_8_H_6_]^+^ [(M+H)−C_8_H_6_−CO]^+^ [(M+H)−C_8_H_6_−CO−H_2_O]^+^ [(M+H)−C_7_H_4_O_5_]^+^	C_15_H_9_O_4_ C_14_H_9_O_3_ C_7_H_5_O_5_ C_6_H_5_O_4_ C_6_H_3_O_3_ C_8_H_7_	253.04919 225.05435 169.01305 141.01811 123.00773 103.04451	[21,25]
Chrysoeriol	C_16_H_12_O_6_	301.07036	0.99	30	[(M+H)−CH_3_]^+•^ [(M+H)−CH_3_−CO]^+•^ [(M+H)−C_9_H_8_O_2_]^+^ [(M+H)−C_7_H_4_O_4_]^+^ [(M+H)−C_8_H_6_O_5_]^+^	C_15_H_10_O_6_ C_14_H_10_O_5_ C_7_H_5_O_4_ C_9_H_9_O_2_ C_8_H_7_O	286.04718 258.05222 153.01855 149.05977 119.04939	[23,24]
Cirsimaritin	C_17_H_14_O_6_	315.08605	0.83	30	[(M+H)−CH_3_]^+•^ [(M+H)−2CH_3_]^+•^ [(M+H)−CH_3_−H_2_O]^+•^ [(M+H)−CH_3_−H_2_O−CO]^+•^ [(M+H)−CH_3_−H_2_O−2CO]^+•^ [(M+H)−C_8_H_6_O]^+^	C_16_H_12_O_6_ C_15_H_9_O_6_ C_16_H_10_O_5_ C_15_H_10_O_4_ C_14_H_10_O_3_ C_9_H_9_O_5_	300.06265 285.03961 282.05215 254.05727 226.06232 197.04445	[23,24]
Dihydrobaicalein	C_15_H_12_O_5_	273.07617	0.41	40	[(M+H)−H_2_O]^+^ [(M+H)−C_2_H_2_O]^+^ [(M+H)−C_8_H_8_]^+^ [(M+H)−C_6_H_6_O_4_]^+^	C_15_H_11_O_4_ C_13_H_11_O_4_ C_7_H_5_O_5_ C_9_H_7_O	255.06465 231.06482 169.01292 131.04906	[25]
Eriodictyol	C_15_H_12_O_6_	289.07105	0.57	20	[(M+H)−H_2_O]^+^ [(M+H)−H_2_O−CO]^+^ [(M+H)−C_6_H_6_O_2_]^+^ [(M+H)−H_2_O−C_6_H_4_O_2_]^+^ [(M+H)−C_8_H_8_O_2_]^+^	C_15_H_11_O_5_ C_14_H_11_O_4_ C_9_H_7_O_4_ C_9_H_7_O_3_ C_7_H_5_O_4_	271.05969 243.05995 179.03374 163.03882 153.01810	[23,24]
Galangin	C_15_H_10_O_5_	271.06051	0.49	30	[(M+H)−H_2_O]^+^ [(M+H)−C_2_H_2_O]^+^ [(M+H)−C_6_H_4_O_2_]^+^ [(M+H)−C_8_H_6_O]^+^ [(M+H)−C_6_H_6_O_3_]^+^	C_15_H_9_O_4_ C_13_H_9_O_4_ C_9_H_7_O_3_ C_7_H_5_O_4_ C_9_H_5_O_2_	253.04956 229.04935 163.03894 153.01825 145.02840	[23,24]
Hesperetin	C_16_H_14_O_6_	303.08620	0.35	20	[(M+H)−H_2_O]^+^ [(M+H)−C_2_H_2_O]^+^ [(M+H)−C_7_H_8_O_2_]^+^ [(M+H)−C_9_H_10_O_2_]^+^ [(M+H)−C_7_H_4_O_4_]^+^	C_16_H_13_O_5_ C_14_H_13_O_5_ C_9_H_7_O_4_ C_7_H_5_O_4_ C_9_H_11_O_2_	285.07529 261.07568 179.03389 153.01828 151.07553	[23,24]
Luteolin	C_15_H_10_O_6_	287.05556	0.92	30	[(M+H)−C_2_H_2_O]^+^ [(M+H)−C_6_H_4_O_2_]^+^ [(M+H)−C_8_H_6_O_2_]^+^ [(M+H)−C_8_H_6_O_3_]^+^	C_13_H_9_O_5_ C_9_H_7_O_4_ C_7_H_5_O_4_ C_7_H_5_O_3_	245.04393 179.03409 153.01822 137.02327	[21,25]
Nepetin	C_16_H_12_O_7_	317.06557	0.19	20	[(M+H)−H_2_O]^+^ [(M+H)−CH_2_O]^+^	C_16_H_11_O_6_ C_15_H_11_O_6_	299.05501 287.05500	[30,33]
Naringenin	C_15_H_12_O_5_	273.07629	0.28	20	[(M+H)−H_2_O]^+^ [(M+H)−C_2_H_2_O]^+^ [(M+H)−C_8_H_8_O]^+^ [(M+H)−C_6_H_6_O]^+^ [(M+H)−3C_2_H_2_O]^+^	C_15_H_11_O_4_ C_13_H_11_O_4_ C_7_H_5_O_4_ C_9_H_7_O_4_ C_9_H_7_O_2_	255.06464 231.06487 153.01810 179.03371 147.04393	[23,24]
Pinocembrin	C_15_H_12_O_4_	257.08132	0.20	20	[(M+H)−H_2_O]^+^ [(M+H)−C_2_H_2_O]^+^ [(M+H)−C_6_H_6_]^+^ [(M+H)−C_8_H_8_]^+^	C_15_H_11_O_3_ C_13_H_11_O_3_ C_9_H_7_O_4_ C_7_H_5_O_4_	239.06998 215.07010 179.03415 153.01816	[21,24]
Quercetin	C_15_H_10_O_7_	303.05047	0.12	20	[(M+H)−H_2_O]^+^ [(M+H)−2CO]^+^ [(M+H)−H_2_O−2CO]^+^ [(M+H)−C_8_H_6_O_3_]^+^	C_15_H_9_O_6_ C_13_H_11_O_5_ C_13_H_9_O_4_ C_7_H_5_O_4_	285.03787 247.06012 229.04951 153.01836	[23,24]
Sakuranetin	C_16_H_14_O_5_	287.09132	0.28	20	[(M+H)−C_2_H_2_O]^+^ [(M+H)−C_6_H_6_O]^+^ [(M+H)−C_8_H_8_O]^+^ [(M+H)−C_7_H_8_O_3_]^+^	C_14_H_13_O_4_ C_10_H_9_O_4_ C_8_H_7_O_4_ C_9_H_7_O_2_	245.08090 193.04971 167.03386 147.04402	[23,24]
Salvigenin	C_18_H_16_O_6_	329.10144	0.52	40	[(M+H)−CH_3_]^+∙^ [(M+H)−CH_3_−H_2_O]^+∙^ [(M+H)−CH_3_−H_2_O−CO]^+∙^	C_17_H_14_O_6_ C_17_H_12_O_5_ C_16_H_12_O_4_	314.07794 296.06750 268.07200	[22,23]
Taxifolin	C_15_H_12_O_7_	305.06548	0.30	30	[(M+H)−H_2_O]^+^ [(M+H)−H_2_O−CO]^+^ [(M+H)−H_2_O−2CO]^+^ [(M+H)−H_2_O−2CO−H_2_O]^+^ [(M+H)−C_8_H_8_O_3_]^+^	C_15_H_11_O_6_ C_14_H_11_O_5_ C_13_H_11_O_4_ C_13_H_9_O_3_ C_7_H_5_O_4_	287.05487 259.06003 231.06512 213.05463 153.01826	[23,24]
Scutellarein	C_15_H_10_O_6_	287.05444	0.57	60	[(M+H)−H_2_O]^+^ [(M+H)−H_2_O−CO]^+^ [(M+H)−C_8_H_6_O]^+^ [(M+H)−C_8_H_6_O−CO]^+^ [(M+H)−C_8_H_6_O−CO−H_2_O]^+^ [(M+H)−C_7_H_4_O_5_]^+^	C_15_H_9_O_5_ C_14_H_9_O_4_ C_7_H_5_O_5_ C_6_H_5_O_4_ C_6_H_3_O_3_ C_8_H_7_O	269.04416 241.04912 169.01280 141.01810 123.00771 119.04921	[25]
Wogonin	C_16_H_12_O_5_	285.07575	0.49	50	[(M+H)−CH_3_]^+•^ [(M+H)−CH_3_−CO]^+•^	C_15_H_10_O_5_ C_14_H_10_O_4_	270.05176 242.05701	[25]
Methyl-apigenin	C_16_H_12_O_5_	285.07687	0.37	30	[(M+H)−CH_3_]^+•^ [(M+H)−CH_3_−CO]^+•^ [(M+H)−C_9_H_8_O]^+^	C_15_H_10_O_5_ C_14_H_10_O_4_ C_7_H_5_O_4_	270.05219 242.05734 153.01848	[23,24]
Methyl-galangin	C_16_H_12_O_5_	285.07563	0.42	30	[(M+H)−CH_3_]^+•^ [(M+H)−CH_3_−CO]^+•^ [(M+H)−CH_3_−CO−HCO]^+•^ [(M+H)−C_9_H_8_O]^+^	C_15_H_10_O_5_ C_14_H_10_O_4_ C_13_H_9_O_3_ C_7_H_5_O_4_	270.05219 242.05734 213.05461 153.01830	[24]
Trimethyl-tricetin	C_18_H_16_O_7_	345.09659	0.80	30	[(M+H)−CH_3_]^+•^ [(M+H)−2CH_3_]^+^ [(M+H)−CH_3_−H_2_O]^+•^ [(M+H)−CH_3_−H_2_O−CO]^+•^ [(M+H)−C_11_H_12_O_3_]^+^	C_17_H_14_O_7_ C_16_H_11_O_7_ C_17_H_12_O_6_ C_16_H_12_O_5_ C_7_H_5_O_4_	330.07339 315.05063 312.06262 284.06769 153.01817	[23,30]

Tentative identification based on comparison with [M^+^] or [M+H]^+^ ions reported in previous studies and the literature for *Scutellaria* genus [22,25], *L. origanoides* chemotypes [20,23,24,31] and *L. alba* chemotypes [26,32]. Tentative identification based on comparison with molecule fragmentation pattern in mass spectra and on databases [33].

**Table 4 molecules-30-04566-t004:** Amounts (mg/g) of flavonoids in extracts prepared in the study.

Name	*S. incarnata*	*S. ventenatti* + *incarnata*	*L. alba* Chemotypes		*L. origanoides* Chemotypes			
	SiUA-2	SviUA	LacUA	LaiUA	LopUA-2	LocUA-1	LocUA-2	LotUA
**Flavonoid glycosides**								
Apigenin-7-glucoside	<LOD	-	<LOD	<LOD	<LOD	0.04 ± 0.0	<LOD	0.62 ± 0.2
Apigenin-7-glucuronide ^a^	-	-	1.5 ± 0.1	2.3 ± 0.2	14 ± 1.0	-	27 ± 1.4	-
Baicalin	15.9 ± 0.0	16 ± 0.0	-	-	-	-	-	-
Chrysoeriol-7-glucuronide	-	-	3.8 ± 0.0	3.5 ± 0.0	-	-	-	-
Chrysoeriol-7-diglucuronide	-	-	4.9 ± 0.1	2.5 ± 0.0	-	-	-	-
Dihydrobaicalein-glucuronide ^a^	23 ± 0.0	<LOD	-	-	-	-	-	-
Eriodictyol-7-glucoside	-	-	-	-	80 ± 2.0	0.36 ± 0.0	89 ± 1.3	3.2 ± 0.3
Eriodictyol-rhamnoside	-	-	-	-	-	0.13 ± 0.0	-	0.32 ± 0.0
Galangin-glucoside ^b^	-	-	-	-	-	<LOD	-	-
Luteolin-7-glucoside	-	-	0.6 ± 0.1	0.68 ± 0.0	49 ± 1.0	0.25 ± 0.0	35 ± 0.1	5.01 ± 0.3
Luteolin-7-glucuronide	-	-	0.8 ± 0.2	1.1 ± 0.3	-	-	-	-
Luteolin-rutinoside	-	-	-	-	-	0.02 ± 0.0	-	0.10 ± 0.0
Quercetin-3-glucoside	-	-	-	-	<LOD	1.5 ± 0.3	<LOD	6.2 ± 0.8
Scutellarin	11.4 ± 0.0	11.4 ± 0.0	-	-	-	-	-	-
Taxifolin-glucoside ^c^	-	-	-	-	-	<LOD	-	<LOD
Tricin-glucuronide	-	-	3.7 ± 0.1	2.8 ± 0.0	-	-	-	-
Tricin-7-diglucuronoside	-	-	5.5 ± 0.1	3.8 ± 0.0	-	-	-	-
Wogonoside	4.7 ± 0.0	4.7 ± 0.0	-	-	-	-	-	-
Phloridzin	-	-	-	-	-	0.01 ± 0.0	-	0.05 ± 0.0
**Flavonoid aglycones**								
Acacetin	-	-	<LOD	<LOD	-	-	-	-
Apigenin	-	-	0.34 ± 0.1	1.2 ± 0.4	-	0.13 ± 0.0	4.3 ± 0.1	0.38 ± 0.0
Baicalein	10.1 ± 0.0	10 ± 0.0	-	-	-	-	-	-
Chrysoeriol	-	-	-	-	-	0.70 ± 0.0	-	2.3 ± 0.1
Cirsimaritin	-	-	0.31 ± 0.0	0.40 ± 0.1	-	0.9 ± 0.0	-	2.9 ± 0.2
Dihydrobaicalein ^d^	4.13 ± 0.007	<LOD	-	-	-	-	-	-
Eriodictyol	-	-	-	-	89 ± 1.00	4.5 ± 0.03	94 ± 4.00	16 ± 2.0
Galangin	-	-	-	-	50 ± 2.00	<LOD	-	<LOD
Hesperetin	-	-	-	-	-	0.17 ± 0.01	<LOD	0.6 ± 0.04
Luteolin	-	-	<LOD	<LOD	7.3 ± 0.20	1.12 ± 0.05	<LOD	3.1 ± 0.14
Naringenin	-	-	<LOD	<LOD	6.0 ± 0.20	1.38 ± 0.05	5 ± 1.00	6 ± 0.2
Nepetin	-	-	<LOD	<LOD	-	-	-	-
Pinocembrin	-	-	-	-	71 ± 2.00	0.03 ± 0.01	<LOD	0.18 ± 0.01
Quercetin	-	-	-	-	<LOD	1.5 ± 0.1	<LOD	8.3 ± 0.6
Sakuranetin	-	-	-	-	<LOD	0.32 ± 0.02	-	0.96 ± 0.04
Salvigenin	-	-	<LOD	<LOD	-	-	-	-
Scutellarein	0.2 ± 0.2	0.2 ± 0.20	-	-	-	-	-	-
Taxifolin	-	-	-	-	4.8 ± 0.10	0.47 ± 0.02	4.7 ± 0.10	1 ± 0.03
Wogonin	15.4 ± 0.03	15 ± 0.03	-	-	-	-	-	-
**Methylated flavonoids**								
Methylapigenin	-	-	-	-	-	0.03 ± 0.01	-	0.07 ± 0.04
Methylgalangin	-	-	-	-	<LOD	<LOD	-	<LOD
Trimethyltricetin	-	-	-	-	-	0.03 ± 0.01	-	0.07 ± 0.01

All extracts were prepared using the UAE technique (Table 1). Extracts from *S. incarnata* (SiUA-2), *S. ventenatti* + *S. incarnata* (SviUA), *L. alba* carvone (LacUA) and citral (LaiUA) chemotypes; and *L. origanoides* phellandrene (LopUA-2), carvacrol (LocUA-1 and LocUA-2) and thymol (LotUA) chemotypes. ^a^ Amounts are expressed as baicalin equivalents; ^b^ amounts are expressed as galangin equivalents; ^c^ amounts are expressed as taxifolin equivalents; ^d^ amounts are expressed as baicalein equivalents. LOD: limit of detection. Apigenin-7-glucoside (LOD = 0.08 mg/L), baicalin (LOD = 0.07 mg/L), galangin (LOD = 0.9 mg/L), quercetin-3-glucoside (LOD = 1.10 mg/L), taxifolin (LOD = 0.08 mg/L), acacetin (LOD = 0.05 mg/L), baicalein (LOD = 0.07 mg/L), luteolin (LOD = 0.06 mg/L), naringenin (LOD = 0.05 mg/L), nepetin (LOD = 0.05 mg/L), quercetin (LOD = 0.05 mg/L), sakuranetin (LOD = 0.06 mg/L), and salvigenin (LOD = 0.06 mg/L).

**Table 5 molecules-30-04566-t005:** Antiviral effect and relative concentrations of flavonoid classes of the extracts.

Extract	Antiviral Effect	Total (mg/g)	Glycoside (%)				Aglycone (%)			Methylated (%)
			A	B	C	D	A	B	C	
ScUA	Strong	277.86	83.4	4.1	0	0.56	10.3	0	0	0
SiUA-1	Strong	492.98	86.1	2.5	0	0	5.0	0	0	0
SviUA	Weak	57.92	55.5	0	0	0	44.5	0	0	0
LacUA	Weak	21.37	97.0	0	0	0	3.0	0	0	0
LaiUA	Weak	18.34	90.7	0	0	0	9.3	0	0	0
SiUA-2	Inactive	84.92	64.8	0	0	0	35.2	0	0	0
LopUA-1	Inactive	27.58	24.7	15	37	0.9	3.9	10.1	8.0	0.4
LopUA-2	Inactive	371.10	17	21.6	0	0	2.0	44.7	14.8	0
LocUA-1	Inactive	13.59	2.3	3.6	11.0	0.07	21.0	47.1	14.5	0.4
LocUA-2	Inactive	259.00	23.9	34.4	0	0	1.7	38.2	1.8	0
LotUA	Inactive	57.44	10	6.1	10.8	0.1	15.1	41.4	16.2	0.2
LopSFE	Inactive	62.96	0	0	0	0	0.2	81.4	11.8	6.5
LocSFE	Inactive	19.85	0	0	0	0	21.4	70.7	6.0	1.9
LotSFE	Inactive	25.90	0	0	0	0	25.8	68.0	0.8	5.4

The total flavonoid content was found to account for 100% of the compounds identified in all extracts, except ScUA (89.5%) and SiUA-1 (93.5%). A: Flavone. B: Flavanone. C: Flavanol + Flavanonol. D: Chalcone. Data are derived from both Table 3 and Appendix A.

**Table 6 molecules-30-04566-t006:** Binding mode predicted for the predominant flavonoids in extracts from *S. coccinea*, *S. incarnata*, *S. ventenatti* + *incarnata*, and *L. alba* to targets. Protein (PDB ID): Cl, clathrin N-terminal domain (2XZG); DENV-2 E, envelope (1OAN); DY, dynamin GTPase domain (2X2E); GX: Gas6-Axl receptor (2C5D).

Name/PubChem CID/Formula	Target: Site	Amino Acids Interacting Through Hydrogen Bonds	kcal/mol
Chrysoeriol-7-diglucuronide 44258206 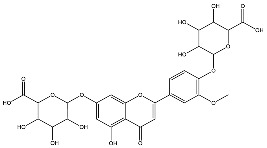	Cl: W-box	Ser34, Thr158, Gln265, Glu268, Arg157.	−10.01 ± 0.27
	DY: G4-switch1	Ser41, Gly60, Lys113, Ala177, Asp180, Leu181, Lys206, Thr214, Asn236, Gln239.	−9.96 ± 0.44
	E: DII (A/B)	Trp206, Thr239, His261, Thr265, Glu269, Ile270.	−8.78 ± 0.37
	GX: Gas6-Lg1	Gln78, Ser304, Gly307, Arg308, Leu309, Gln341, Ile426, Phe428, His429.	−8.69 ± 0.18
Baicalin 64982 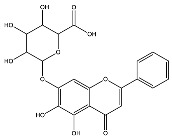	DY: G4-switch1	Ser41, Gly60, Ser61, Ala177, Asp180, Lys206, Thr214.	−9.67 ± 0.17
	Cl: W-box	Glu33, Lys112, Asn155, Arg157, The158, Met264, Gln265, Thr306	−9.33 ± 0.49
	GX: Gas6-Lg1/Lg2	Met644, Thr645, Asp654, Leu655, Ala667.	−7.96 ± 0.18
	E: DII/DI	Arg2, Glu44, Ile46, Asp98.	−7.96 ± 0.22
Tricin-7-diglucuronoside 131752191 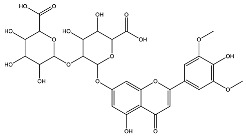	Cl: W-box	Lys112, Arg157, Ser200, Phe201, Met264, Gln265, Ile266, Glu268, Thr306, Gly314.	−9.45 ± 0.17
	E: DII (A/B)	Leu65, Asp203, Lys204, Ala205, Val252, His261, Glu269.	−8.29 ± 0.21
	DY: P-loop	Gln33, Arg67, Arg107, Gln117, Asn126, Asp130, Leu131, Lys166.	−7.88 ± 0.14
	GX: Gas6-Lg1/Axl-Ig2	Gln78, Ser302, Gly303, Arg308, Leu309, Gln341, Ile426, Phe428, His429.	−8.05 ± 0.20
Scutellarin 185617 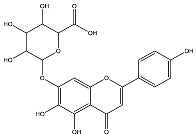	Cl: W-box	Trp111, Lys112, Trp113, Arg157, Thr158, Met264, Gln265, Thr306.	−9.28 ± 0.39
	DY: G4-switch1	Ser61, Asp180, Leu181, Lys206, Thr214.	−8.31 ± 0.16
	E: DII (A/B)	Asp203, Lys204, Ala205, Thr262, Leu264, Thr265, Glu269.	−8.13 ± 0.32
	GX: Gas6-Lg1/Lg2	Glu331, Asn438, Arg476, Gly477, Asp654, Glu657, Ala667.	−7.80 ± 0.17
Apigenin-7-glucuronide 5319484 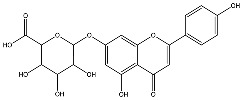	Cl: W-box	Lys112, Thr158, Gln265, Thr306	−9.17 ± 0.27
	DY: G4-switch1	Ser41, Ala42, Gly43, Lys44, Ser45, Ser46, Gly60, Val64, Thr65, Gln239.	−8.70 ± 0.16
	E: DII (A/B)	Asp203, Lys204, Leu264, Thr262, Glu269.	−8.22 ± 0.17
	GX: Gas6-Lg1/Lg2	Phe328, Asn438, Arg476, Gly477, Asp654, Glu657, Ala667, His668.	−7.82 ± 0.17
Luteolin-7-glucuronide 5280601 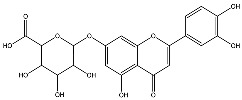	Cl: W-box	Glu33, Ser70, The109, Lys112, Ile153	−9.10 ± 0.26
	DY: G4-switch1	Ser41, Ala42, Gly43, Lys44, Ser46, Arg59, Gly60, Val64, Thr65, Lys206, Leu207, Asn236.	−8.52 ± 0.18
	E: DII (A/B)	Lys204, Ala205, Thr265, Ala267, Glu269	−8.41 ± 0.15
	GX: Gas6-Lg2	Gly477, Arg514, Asp654, Ala667.	−8.25 ± 0.21
Tricin-glucuronide 101939793 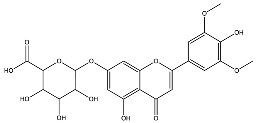	Cl: W-box	Lys112, Arg157, Ser200, Phe201, Met264, Gln265, Ile266, Glu268, Thr306, Gly314.	−9.45 ± 0.17
	GX: Gas6-Lg1/Lg2	Gln78, Ser302, Gly303, Arg308, Leu309, Gln341, Ile426, Phe428, His429.	−8.05 ± 0.20
	E: DII (A/B)	Leu65, Asp203, Lys204, Ala205, Val252, His261, Glu269, Trp206.	−8.29 ± 0.20
	DY: P-loop	Gln117, Asn126, Asp130, Leu131, Gln33, Arg67, Arg107, Lys166.	−7.88 ± 0.14
Dihydrobaicalein-glucuronide 14135324 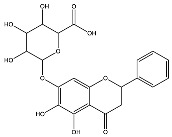	Cl: W-box	Ser28, Ser70, Thr109, Gln152, Ile153.	−8.90 ± 0.27
	DY: P-loop	Gln17, Asn26, Asn121.	−8.41 ± 0.20
	GX: Gas6-Lg2	Gln78, Arg308.	−7.99 ± 0.20
	E: DI/DII	Gly102, Asn103, His149, Gly152, Asn153, Asp154, Thr155.	−7.72 ± 0.34
Isocarthamidin-glucuronide 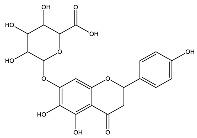	DY: switch2-P	Lys142, Glu153, Lys188.	−8.87 ± 0.26
	Cl: W-box	Trp113, Arg157, Met264, Gln265, Glu268, Thr30.	−8.78 ± 0.35
	GX: Gas6-Lg1/Lg2	Arg476, Gly477, Ser478, Arg514, Thr645.	−8.46 ± 0.18
	E: DI/DII	Asn103, Val151, Asp154, Thr155, His244, Lys246.	−7.97 ± 0.24
Chrysoeriol-7-glucuronide 14630700 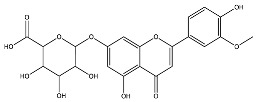	DY: G4-switch1	Ser41, Val64, Thr65, Ser179, Asp180, Lys206, Thr214, Asp215.	−8.85 ± 0.43
	Cl: W-box	Met32, Asn155, Gln265, Thr306, Glu301.	−8.85 ± 0.27
	E: DII (A/B)	Lys204, Trp206, Thr262, Leu264, Thr265, Glu269.	−8.37 ± 0.16
	GX: Gas6-Lg1/Lg2	Lys290, Arg467, Ser663, Asp664.	−7.83 ± 0.16
Luteolin-7-glucoside 5280637 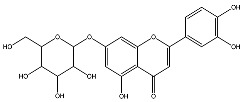	Cl: W-box	Ser70, Thr109, Lys112, Ile153.	−8.54 ± 0.26
	GX: Gas6-Lg2	Ser478, Arg514, Cys643, Asp654, His668.	−8.18 ± 0.19
	DY: G4-switch1	Gly60, Ala177, Lys206, Leu209, Asp211, Asp236.	−8.00 ± 0.24
	E: DII (A/B)	Lys204, Ala205, Thr262, Thr265.	−7.98 ± 0.17
Wogonoside 3084961 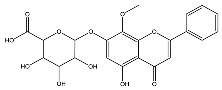	Cl: W-box	Arg157, Thr158, Ala160, Gln162, Phe201, Gln203, Glu268.	−8.51 ± 0.20
	E: DII (A/B)	Trp206, Gln256, Gly258, Leu264, Thr265, Glu269.	−7.94 ± 0.24
	DY: G4-switch1	Arg67, His85, Arg107, Asn121.	−7.69 ± 0.19
	GX: Gas6-Lg1/Lg2	Gln78, Leu309, Arg310, Gln341.	−7.69 ± 0.12
Baicalein 5281605 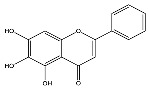	DY: switch2-P	Thr141, Glu153, Arg157, Asp185.	−8.26 ± 0.36
	Cl: W-box	Asn155, Arg157, Val262, Met264, Gln265, Thr306.	−8.08 ± 0.47
	E: βOG pocket	His27, Thr48, Gly281, His282.	−7.92 ± 0.76
Dihydrobaicalein 9816931 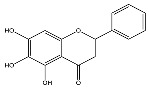	DY: switch2-P	Thr141, Glu153, Arg157, Asp185.	−8.25 ± 0.40
	Cl: W-box	Ser70, Lys83, Trp111.	7.97 ± 0.45
	E: βOG pocket	Leu25, His27, Thr48, Gly281, His282.	−7.84 ± 0.80
Wogonin 5281703 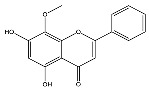	E: βOG pocket	Thr48, Tyr137, Gly281, His281.	−7.66 ± 0.80

Both apigenin-7-glucoronide and luteolin-7-glucoside were found in extracts from *L. origanoides*.

## Data Availability

Data are contained within the article and Appendix A.

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
