# Peer review of "In Vitro and In Silico Analyses Explore the Role of Flavonoid Classes in the Antiviral Activity of Plant Extracts Against the Dengue Virus"

_molecules, 2025, doi:10.3390/molecules30234566_

Round 1

Reviewer 1 Report

Comments and Suggestions for Authors

This research  employs in vitro and in silico analytical methods to explore the association between flavonoids in plant extracts derived from Colombian cultivated medicinal plants and their anti-dengue virus (DENV) activity. The manuscript is data-rich, and the conclusions are largely reliable, though certain aspects require further clarification and refinement.

Comments for this manuscript:

1. For the keywords section, it is recommended to add plant names as keywords.

2. In the text, the phrase 'Scutellaria, L. alba, and L. origanoides' appears multiple times. The first element is a genus name, while the latter two are species names, creating inconsistency. It is recommended to revise this.

3. There is a formatting issue in Table 1. Why are both the plant name 'Scutellaria coccinea Kunth' and the Voucher name 'UIS219784' in bold, with underlines beneath them?

4. Figure 1: Because the number of extracts involved is large, and the differences between markers are small, the readability of this figure is poor. It is suggested to modify the presentation form, such as using different colors? Additionally, please add concentration units to the x-axis of the four figures above.

5. In the cytotoxicity assay, two different methods were adopted. What was the rationale for this? Additionally, DMSO is a common organic solvent used in bioactivity screening. In this experiment, analyzing the toxicity of different DMSO concentrations should aim to evaluate the solvent's impact on the experiment, rather than serving as a positive control.

6. Table 2, 3 and 5: Has formatting errors similar to those in Table 1.

7. Table 3: Formula for Apigenin-7-glucoside was wrong (C2120O10).

8. In the article, plant Latin names appear multiple times without italics. Please correct this.

9. The entire article involves 14 extracts from five plant species. Corresponding extract codes are assigned based on the different plants and extraction methods, but these codes lack strong logical consistency, which hampers the overall readability of the paper and makes confusion likely. It is suggested to revise the coding system to facilitate reader identification.

10. Line 201, 204, 210, and 211: The data involved are inconsistent with those in Table 5. Please carefully check all relevant data throughout the text to ensure data accuracy.

11. By citing references, briefly introduce Kohonen's self-organized map algorithm and what its clustering principle is.

12. Table 6: The chemical structure for Chrysoeriol-7-diglucoronide (the first one) was wrong. Additionally, the chemical structures of the compounds in the table are not clear enough.

13. Line 355: "flavonoids constituted over 100% of their content",  Do the extraction methods adopted in this article enable the flavonoid content in the extracts to reach over 100%?

Comments on the Quality of English Language

No other comments.

Reviewer 2 Report

Comments and Suggestions for Authors

Dengue virus (DENV) is a serious global health problem. Despite its widespread prevalence -especially in tropical and subtropical regions - there is still no specific antiviral treatment available, and current treatment focuses primarily on symptom relief and supportive care. Therefore, exploring new therapeutic alternatives, including natural compounds and plant extracts with potential antiviral activity, is crucial to improving dengue prevention and treatment strategies. Therefore, Sindi Alejandra Velandia's work addresses the ongoing need for effective methods to combat dengue virus.

To ensure the high quality of your manuscript, please address the following issues:

According to the iThenticate report, sections 4.1-4.7 show a high degree of similarity to previously published works and should be paraphrased.

Keywords - consider expanding with “in silico analyses” and “antiviral activity.”

Introduction:

Please expand the description related to the Dengue virus. In particular, it would be valuable to address the following questions:

  • How is the disease transmitted?
  • In which regions, apart from the Americas, is the Dengue virus found?
  • What types of Dengue virus have been confirmed?
  • What are the mortality statistics?

Lines 69-72: please move the following text into the first paragraph concerning the Dengue virus:

“Dengue has been identified as a significant and persistent public health concern in Colombia, which is among one of the most affected countries in the Americas. From 2012 to 2020, annual incidence rate ranged from 90.7 to 476.2 per 100,000 population, with the most recent outbreak reported in 2019 (465.9 per 100,000 population) [25].”

Lines 53-55: please expand the description with information from references [10,11], and then introduce the text from lines 72-76:

“Traditional Colombian herbal remedies are widely used to alleviate dengue symptoms, although few plants have been systematically studied for their therapeutic potential [26]. Research on the anti-DENV activity of plant extracts contributes to discovering alternative therapies to prevent severe dengue.”

Lines 76-79: please add information about the country of origin of the raw materials.

Materials and Methods:

4.3 Extracts

  • Plant material: please specify in parentheses what raw materials were used. Missing details include the harvesting period of the material, whether mass loss after drying was measured, the equipment used for grinding, and whether the material was sieved to standardize particle size.
  • Line 445: “Fourteen extracts were included in the analysis of their anti-DENV effect” - on what basis do the authors make this statement? A precise justification is missing here.
  • There is a lot of confusion in the content. The authors made some of the extracts as part of previous research, and some as part of the current one. It would be necessary to prepare a table for 14 extracts and indicate precisely what raw material / part of it was used, what technique was used, preparation parameters + literature reference if it was described earlier. Were the extraction parameters optimized? This is missing from the manuscript.

4.7 Molecular docking analysis

  • Please provide the Protein Data Bank website and the date of access.
  • Structures of flavonoids: list all the structures used, including their PubChem CIDs in parentheses.
  • Lines 528-530: briefly describe the preparation procedure here.

4.8 Statistical analysis

  • Matlab - please include an appropriate reference.

Results

  • Figures 1 and 2: image quality is poor; font size and overall figure size should be increased, as the current version is difficult to read.
  • Tables 2 and 4: if the error values are given to one decimal place, the CPE and ICâ‚…â‚€ values should also be provided to one decimal place, for consistency.
  • Figure 3: font size is too small, reducing legibility.
  • Figure 5: the close-up views of chrysoeriol-7-diglucuronide and norwogonin have unreadable amino acid labels. The figures are of very low quality in their current form and should be improved. The figure legend should also include an explanation of the meaning of blue, red, gray, and yellow colors.
  • Figures 6-8: similarly to Figure 5, amino acid labels in close-up views are unreadable. The figures are of poor quality and must be improved.
  • Table 6: the quality of the structural graphics is low. Please improve the structures to meet DPI quality guidelines for figure presentation.

Reviewer 3 Report

Comments and Suggestions for Authors

The paper is interesting, well written and it reports the outcome of a lot of work. I have just one point: 

  1. I am unable to find in the manuscript the way the compounds were quantified by hplc. How was it done? Did the authors used a reference compound to obtained the amounts of compounds in mg/g? In this case are the numbers expressed as "equivalent to ..." or similar ways? This detail should be added to the experimental section.

Round 2

Reviewer 2 Report

Comments and Suggestions for Authors

The Authors responded to all comments. The Authors made changes to the manuscript, improving its quality. I accept the manuscript in its current form.